# Trends in vertical wind velocity variability reveal cloud microphysical feedback

**Donifan Barahona** [1] ✉, **Katherine H. Breen**[1,2], **Derek Ngo**[3], **Flor Vanessa Maciel**[3,4], **Ryan Patnaude**[3,5] **& Minghui Diao** [3]

By controlling supersaturation vertical air motion influences how aerosols activate into cloud droplets and ice crystals. This effect is difficult to represent accurately in atmospheric models as they cannot typically resolve the sub-kilometer scale component of wind motion, however it can be addressed by machine learning. Here we apply a generative technique combining storm-resolving simulations, observational and climate reanalysis data, to predict the spatial standard deviation in vertical wind velocity, $\sigma_W$. This analysis reveals significant trends in $\sigma_W$, reaching up to 1%yr$^{-1}$ in low and mid-level oceanic regions, indicating enhanced atmospheric turbulence. We attribute these trends to global shifts in water vapor, temperature and convection, suggesting a feedback connection between enhanced warming and turbulence, which in turn has a microphysical effect through the activation of more cloud hydrometeors. Focusing on low level clouds this effect has had a radiative impact of about −0.10 ± 0.04Wm$^{-2}$ since the year 1900, slightly mitigating greenhouse warming.

Vertical wind motion drives cloud formation by inducing water vapor supersaturation in air parcels, leading to the activation of aerosol particles into cloud droplets and ice crystals[1]. The characteristic scale of these processes ranges from meters to a few kilometers, whereas modern global atmospheric models, with typical spatial resolutions between 10 km and 100 km, cannot fully capture these small-scale phenomena. As a result, vertical transport and cloud formation processes are classified as subgrid-scale and are primarily represented through physical parameterizations in global circulation models (GCMs). A key component of these parameterizations is the distribution of vertical wind velocity ($W$) within each grid cell[2]. While only the mean $W$ can be directly resolved, the moments of the subgrid distribution, particularly the standard deviation ($\sigma_W$), must be estimated from the coarse atmospheric state. Since variability in $W$ produces a spectrum of supersaturation, $\sigma_W$ plays a crucial role in determining grid-mean cloud particle formation rates[1,2]. Additionally, as it reflects the amplitude of vertical wind fluctuations, $\sigma_W$ serves as an indicator of turbulence[3] and gravity wave activity[4].

Vertical wind variability is difficult to parameterize accurately in global atmospheric models and is typically treated using either approximate solutions to the equations of motion at the subgrid scale[5] or empirical parameterizations[6–10] designed to represent specific processes such as turbulence, transport, and cloud microphysics. The diversity of these approaches introduces uncertainty in climate and weather predictions. Although observational data on $\sigma_W$ exists from field campaigns and ground-based measurements[11–13], it is typically localized and episodic. Large Eddy Simulations have been used to develop parameterizations of $\sigma_W$, mostly focusing on stratocumulus regimes[14]. These are however limited in domain and it is not clear to what extent their results are applicable across the wide diversity of meteorological environments present in the atmosphere. Currently, no global dataset exists to constrain $\sigma_W$ in atmospheric models. Since the representation of cloud formation depends strongly on $\sigma_W$, this limitation may impact estimates of cloudiness and cloud feedback[2,15].

In the past decade, the ability of atmospheric models to represent weather patterns has improved thanks to advances in computing

[1]Global Modeling and Assimilation Office, NASA Goddard Space Flight Center, Greenbelt, MD, USA. [2]Morgan State University, Baltimore, MD, USA. [3]Department of Meteorology and Climate Science, San Jose State University, San Jose, CA, USA. [4]Department of Atmospheric and Oceanic Sciences, University of California, Los Angeles, CA, USA. [5]Department of Atmospheric Science, Colorado State University, Fort Collins, CO, USA. ✉ e-mail: donifan.o.barahona@nasa.gov

power, with many achieving kilometer-scale resolution (i.e., with spatial grids between 3 km and 7 km)[16], known as global storm resolving models (GSRMs). Compared to traditional GCMs these simulations can better capture convection, vertical transport and cloud microphysics[17]. However, even at such scale GSRMs still misrepresent high frequency gravity waves[18] and turbulence[14], for which even finer resolution may be required[19–21]. The GSRM simulations, and the extensive data they produce, however represent a wealth of information that can be used to understand and model $\sigma_W$, particularly when combined with observational datasets. Artificial intelligence techniques and data-driven algorithms like deep learning are well-suited for this task. Unlike traditional numerical simulations or theories, these algorithms learn from data to make predictions.

Using deep learning to integrate model results with experimental measurements of $\sigma_W$ provides a way to correct for the physical approximations in simulated data. A recent study[21] introduced a model of this kind, termed Wnet, designed to predict $\sigma_W$ using the coarse (spatial resolution $\Delta x \sim 0.25°-1°$) meteorological state resolved by typical climate models. Wnet was developed in two stages: first, it was trained on results from GSRM simulations ($\Delta x \sim 6$ km), then refined using ground-based observations ($\Delta x \sim 30-300$ m) from sites worldwide. To account for experimental uncertainty in the observations, the refinement process is based on generative adversarial networks[22,23]. This approach introduces an auxiliary network, called the discriminator, which learns to reject noise by identifying structured features in real data. The discriminator guides the training process by penalizing Wnet when it produces noisy output, forcing the model to distinguish meaningful patterns from unstructured noise. As a result, experimental error is gradually filtered out, aligning Wnet's predictions with the observed distribution and converging to a clean approximation of the true data. This represents an advancement in the parameterization of $\sigma_W$, as $W$ retrievals are inherently challenging and often associated with substantial uncertainty[11,12].

The global simulation data inform Wnet about the spatial distribution of $\sigma_W$, enabling the neural network to extrapolate across different scenarios and cloud formation regimes. The refinement step corrects for unresolved variability in the simulations. This synergy between high-resolution simulations and observational data enables Wnet to predict $\sigma_W$ across a wide range of atmospheric conditions[21]. The Wnet model takes as input coarse-resolution meteorological and dynamical factors, including winds, temperature, water concentration and phase, and simplified turbulence metrics, such as the Richardson number and scalar diffusivity, to estimate unresolved variability in vertical motion, bridging large-scale and small-scale dynamics.

Wnet can be run online, applied to prior simulations, or used with weather forecast and reanalysis products. The latter is particularly beneficial, as climate reanalyses rely on traditional assimilation techniques to incorporate millions of observations, reconstructing the observed meteorological state[24] and providing unique historical, global atmospheric data spanning multiple decades. Since Wnet was built to use variables that can be provided by reanalyses, it is possible to reconstruct $\sigma_W$ globally using their output. Analyzing long-term trends in such datasets help us to understand how changes in large-scale weather patterns affect small-scale dynamical processes, and how these in turn, feedback on climate. We will present an analysis showing significant trends in $\sigma_W$ emerging over recent decades, with implications not only for cloud formation but also climate change assessments, and weather and turbulence forecast.

## Results

We used meteorological fields from two widely employed reanalyses, MERRA-2[25] and ERA5[26], as input to the Wnet model to create three-dimensional global climatologies of $\sigma_W$, every 3 h from 1980 to 2022. While both ERA5 and MERRA2 rely heavily on observed data, they differ in assimilation methods and in their underlying modeling

approach, resulting in differences in the global distribution of clouds, water vapor, convection and winds[27–29]. Since MERRA-2 was used to provide the meteorological state during Wnet training (when refining against observations), using ERA5 allows us to assess how sensitive $\sigma_W$ is to the choice of input data.

Figure 1 displays the global mean spatial distribution of $\sigma_W$, at a half-degree horizontal resolution as the average of the prediction from both reanalyses. There is a broad range of $\sigma_W$ values, peaking near mountain ranges and within the inter-tropical convergence zone (ITCZ). In the mid-latitudes, $\sigma_W$ tends to decrease with height as turbulence weakens and the effects of surface roughness and boundary layer mixing diminish[30]. In contrast, deep convection in the tropics generates high-frequency gravity waves through latent heat release in the middle and upper troposphere, which tend to enhance $\sigma_W$[31]. Additional factors such as boundary layer mixing and frontal systems contribute to variability[32], producing regions of elevated $\sigma_W$ near the surface (see Fig. 1 at 900 hPa).

These features are also evident when using MERRA-2 (Fig. S4) and ERA5 (Fig. S6) separately as input to Wnet. Both reanalyses produce similar global distributions of $\sigma_W$, with high values near orographic features and tropical convection. Some differences appear at 500 hPa, where ERA5 shows a slightly lower global average $\sigma_W$ compared to MERRA-2. This is likely due to more active convection in MERRA-2, which transports water vapor and detrains condensate at higher altitudes[28]. The Wnet model is sensitive to such differences, as the presence of clouds leads to latent heat release that perturbs the local environment and enhances turbulence, increasing $\sigma_W$. Despite these variations, both datasets yield similar spatial patterns of $\sigma_W$, providing confidence that the generative method used to train Wnet effectively filters out error producing an unbiased estimate of $\sigma_W$.

## Validation of the $\sigma_W$ datasets

In situ measurements and ground-based observations are used to validate the climatologies of $\sigma_W$. Since $\sigma_W$ is generally not measured directly, it is derived from high-frequency (typically 1–10 Hz) measurements of $W$[11,13], corresponding to spatial scales of roughly 30–300 m, which encompass most vertical motions relevant for cloud formation. To calculate $\sigma_W$, the high-resolution $W$ fields are used to compute standard deviations over spatial windows corresponding to a characteristic scale of ~50–100 km, representative of variability on scales comparable to a typical GCM grid cell. For aircraft measurements, $\sigma_W$ was calculated using a 430 s averaging window, representing a horizontal scale of $\Delta x \sim 100$ km[13], covering both in-cloud and clear-sky regions. For ground-based retrievals, the characteristic time scale depends on the mean horizontal wind at each site and ranges from 15 to 30 min, corresponding to $\Delta x \sim 50$ km[21]. Observations were collected from multiple locations worldwide, spanning a wide range of cloud regimes (cirrus, stratocumulus, altocumulus, convective) and employing various measurement techniques for $W$, providing independent validation of $\sigma_W$.

We compare the estimates of $\sigma_W$ against in situ measurements from over 1000 h of research flight data, collected during 12 field campaigns across North and South America and the Tropical Pacific, covering much of the western hemisphere (Table S1) [cf. Table 1[13]]. All field data were thoroughly quality-controlled to ensure reliability[13,20]. For each campaign, the $\sigma_W$ datasets were interpolated and compared to measured values along each flight track. Since these campaigns were designed to study high clouds, the data represent free-tropospheric conditions not included in the training set, providing a test of Wnet's ability to generalize beyond its training data. Figure 2 shows that, for most campaigns, the agreement between predicted and measured $\sigma_W$ is within 0.1 ms⁻¹, well inside the typical experimental uncertainty for vertical wind velocity measurements[33]. Wnet also reproduces much of the observed variability in $\sigma_W$, though it is

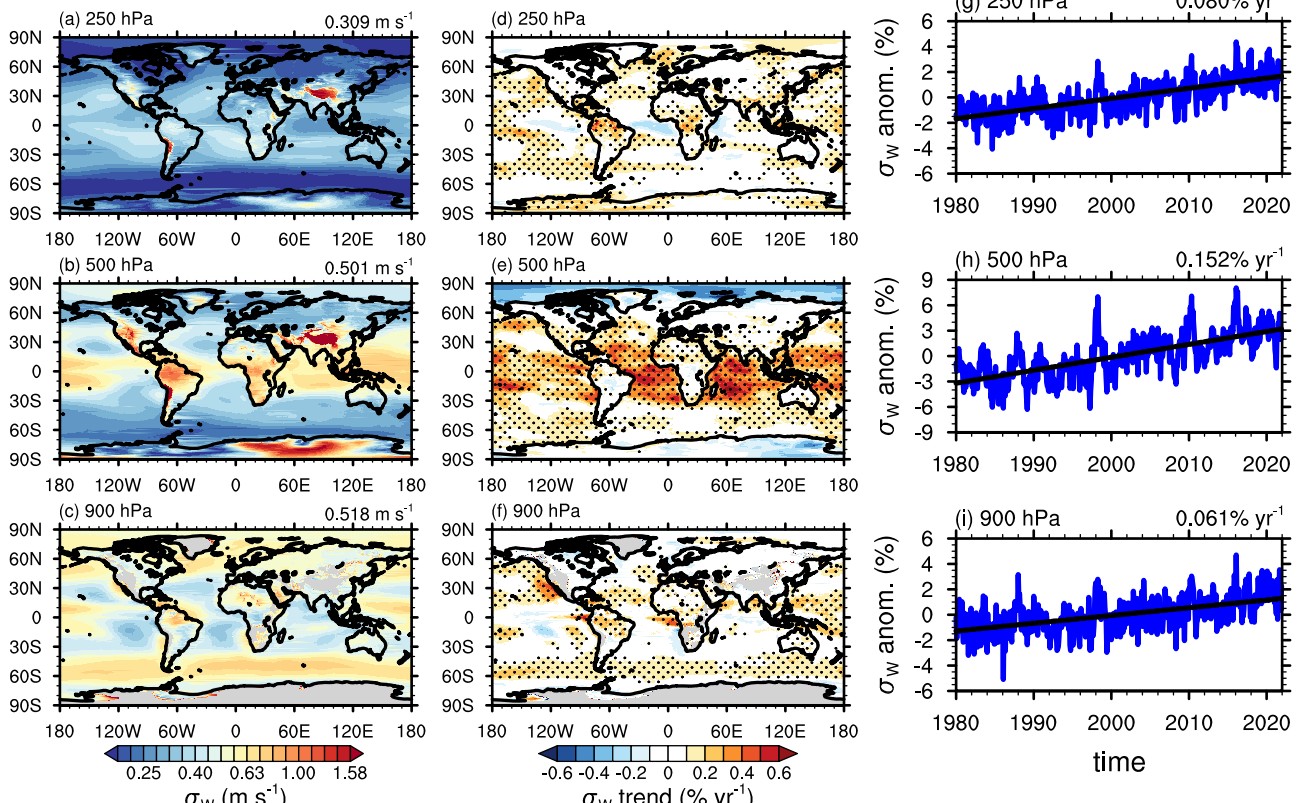

**Fig. 1 | Global trends in the standard deviation in the vertical wind velocity ($\sigma_W$) obtained from applying the Wnet model to reanalysis data. a–c** The annual mean $\sigma_W$ for the period 1980−2022. **d–f** The deseasonalized linear trend, relative to the long-term mean, derived from monthly means. Stippling highlights statistical significance at the 95% level. **g–i** The evolution of the global average trend at each level. The long-term mean is displayed at the top right corner.

underestimated in two campaigns (ATTREX and POSIDON) where conditions were strongly influenced by convection. These environments were underrepresented in the training dataset[21] and tend to have a larger experimental error (-0.3 ms⁻¹)[13]. Using ERA5 tends to produce a slightly narrower range of $\sigma_W$ values compared to MERRA-2 and the observations, indicating that Wnet driven by ERA5 predicts less extreme values, also evident in Figs. S4 and S6. As ERA5 ($\Delta x \sim 0.25°$) has finer spatial resolution than MERRA-2 ($\Delta x \sim 0.5°$) it may result in a slightly larger fraction of resolved variability and a smaller subgrid-scale $\sigma_W$.

We also compared the predicted $\sigma_W$ with remote sensing retrievals from 11 ground-based sites worldwide, as listed in Table S2, encompassing more than 100 years of high-frequency $\sigma_W$ data. As with the aircraft measurements, the predicted $\sigma_W$ generally falls within the typical experimental error, as shown in Fig. 2. Measurements taken within cirrus clouds using radar are restricted to cloudy regions and are about an order of magnitude lower than near the surface, within the planetary boundary layer (PBL). Wnet represents this wide variation in $\sigma_W$ performing well across diverse environments. The best agreement is obtained at the cirrus sites[12] and in sites typical of low-level cloud decks like ENA[34], NSA[35] and ASI[36]. Larger discrepancies occur at Tropical sites dominated by convection like MAO[37] and TWP[38], likely due to the limited representation of convective regimes in the training dataset[21]. Wnet, driven by MERRA-2 meteorology, was partially trained using these data[21] and as before represents a slightly wider variability in $\sigma_W$ than ERA5, although both still lie within the observed range and have means close to the observations.

Remote sensing retrievals mainly sampled cirrus clouds and the PBL, while aircraft measurements spanned a wider range of altitudes, primarily within stratocumulus and altocumulus

environments. Some campaigns, such as ATTREX[39], also captured high-altitude clouds with temperatures below 200 K. Figure 2 shows that the derived climatologies reproduce the observed $\sigma_W$ well across these regimes. Neither the in situ aircraft observations nor the ERA5 data were used in developing Wnet, providing an independent validation. Wnet's strong performance across diverse conditions and measurement techniques indicates that the training algorithm effectively reduced experimental errors and captured the physical relationships between the coarse atmospheric state and $\sigma_W$.

## Long-term trends in $\sigma_W$

The long-term climatologies of $\sigma_W$ derived from reanalysis data offer a unique opportunity to examine its multi-decadal behavior in relation to changes in meteorological conditions. Wind variability is influenced by factors like convection and orography, which create gravity waves and turbulence[32,40]. Figure 1 shows the linear trend in the $\sigma_W$ anomaly for each grid cell calculated from monthly averages over the 1980−2022 period. On average, there is a statistically significant, positive trend in $\sigma_W$ on the order of 0.1% per year (equivalent to a global increase of 0.5−1 cm s⁻¹ per decade). There is, however, substantial spatial variation, as positive trends are generally observed in the Tropics and midlatitudes, while negative trends are seen in the Arctic and Antarctic, and in some subtropical regions like the west coasts of South America and South Africa. These trends also vary with altitude, with the highest values reaching ~1%yr⁻¹ in the middle troposphere (around 500 hPa) in the Tropics and the most negative in the Arctic (approximately −0.5%yr⁻¹). Despite negative trends in some areas, the overall global average indicates an increase in $\sigma_W$ across most of the troposphere over the past four decades. The statistically

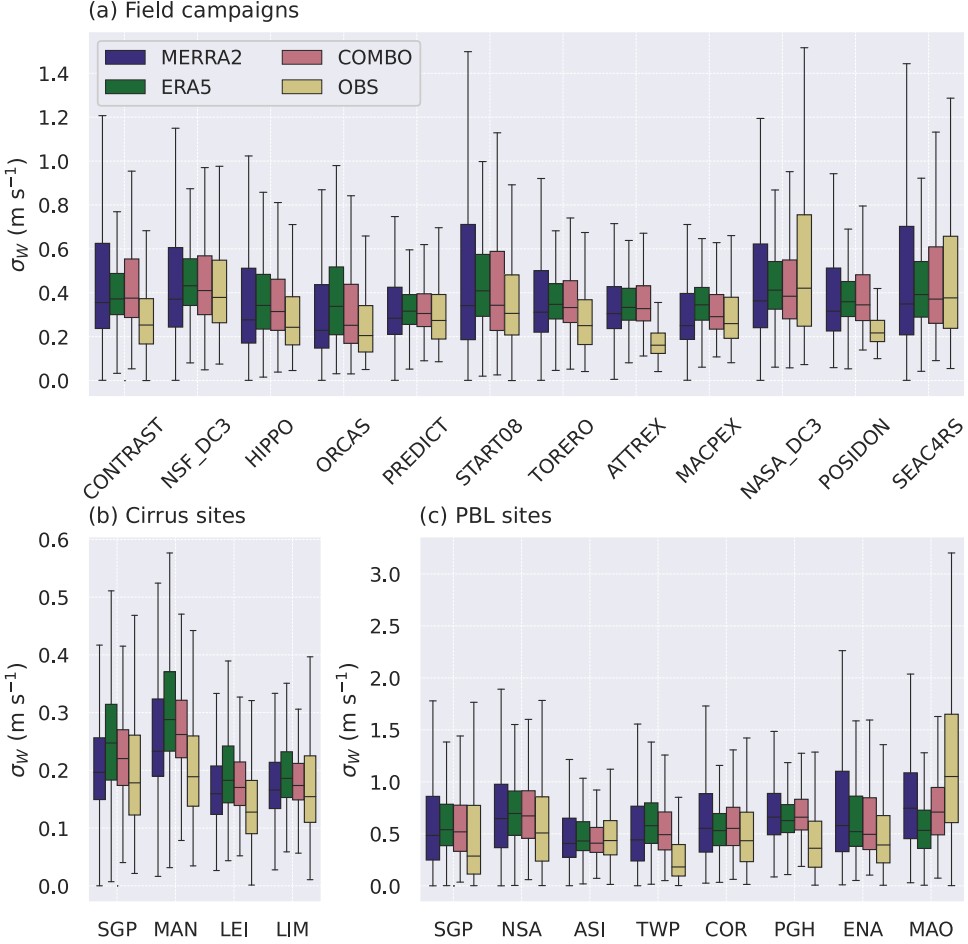

**Fig. 2 | Validation of the standard deviation in the vertical wind velocity ($\sigma_W$) against observations (OBS).** Three data sources were used to drive the Wnet model: ERA5, MERRA-2, and their combination (COMBO). Observations include (**a**) aircraft measurements in cirrus clouds, (**b**) ground-based remote sensing in cirrus clouds, and (**c**) ground-based remote sensing in the planetary boundary layer (PBL). For clarity the MAO data was scaled by a factor of 0.7. The location and time periods for each field campaign and ground site are summarized in Tables S1 and S2.

significant trends highlight areas where atmospheric dynamics are evolving, possibly linked to changing climate patterns.

Significant anomalies in $\sigma_W$ are observed around 500 hPa, where some areas in the Tropics experienced up to a 30% increase between 1980 and 2020 (see Figs. S5 and S7). Conversely, high latitudes beyond 60S and 60N saw almost a 20% decrease over the same period. Near the surface (900 hpa) regions characterized by persistent stratocumulus, off the coasts of North and South America and east Africa, as well as the "roaring 40s" (40S-50S) also saw a significant increase in $\sigma_W$ between 1980 and 2020, although some regions near the ITCZ and west Africa showed negative trends. Similar patterns are observed at 250 hPa, although at this level, $\sigma_W$ generally increased in most oceanic regions. Anomalies tended to be higher over the ocean compared to land, except for South America at 900 hPa and Australia at 250 hPa. Positive anomalies also developed earlier at 250 hPa (starting after 1990) than at 900 hPa (starting after 2000), suggesting that different mechanisms may be responsible for the observed trends at different levels.

Similar patterns emerge for the ERA5 and MERRA-2 reanalyses, showing pronounced trends in the tropical Pacific (Figs. S4 and S6), particularly at 500 hPa and near the surface over the Indian Ocean and equatorial Pacific, the latter likely linked to boundary-layer turbulence. Negative trends in ERA5 are more pronounced in the Arctic at 500 hPa and in the western Pacific at 900 hPa, compared to MERRA-2. Consequently, the overall global trend in $\sigma_W$ tends to be weaker in ERA5 than

in MERRA-2, although it remains statistically significant across most regions, with the highest values occurring in the tropical lower troposphere. These differences are expected, as the evolution of the meteorological state, particularly water vapor and cloudiness, differs significantly between the MERRA-2 and ERA5 reanalyses[41], which influences how $\sigma_W$ evolves when Wnet is driven by each dataset. Despite differences in the absolute magnitude of the trends derived from ERA5 and MERRA-2, both datasets consistently indicate a net global increase in $\sigma_W$ over recent decades. This strongly supports the hypothesis of a global intensification of subgrid-scale variability, with statistically significant changes concentrated in regions of active weather systems. These findings suggest that the evolution of large-scale climatic patterns influence subgrid scale variability, potentially impacting the long-term evolution of clouds.

**Origin of the observed trends in $\sigma_w$**

We use Shapley Additive explanations, SHAP[42], to investigate the contribution of each input variable to the trends depicted in Fig. 1. SHAP is an interpretable machine learning technique based on Shapley values, which indicate the average contribution of each input to the overall deviation in $\sigma_W$ from its average value. The Wnet model employs 14 scalar inputs, 10 at the grid level and 4 from the surface level at the same location. Combining surface variables with each grid-level input allows Wnet to account for the effects of orographic features and convection initiation on $\sigma_W$, while remaining independent of

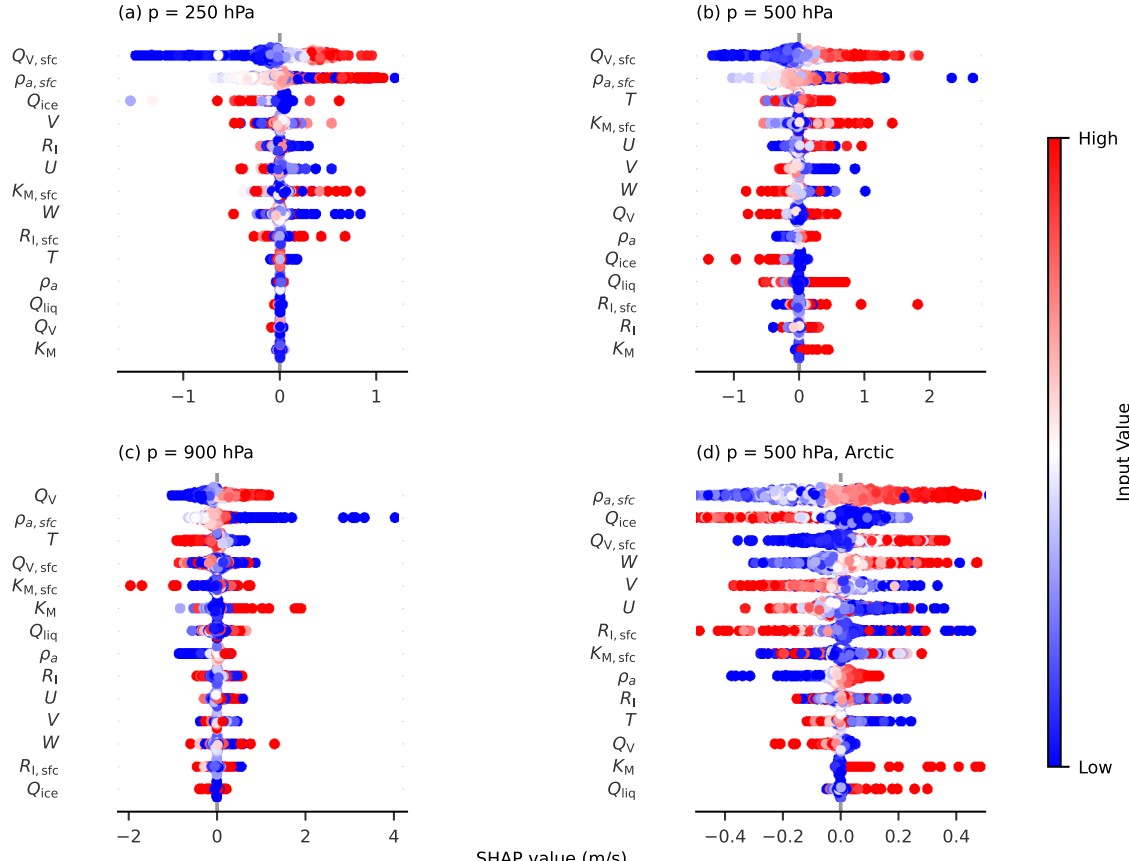

**Fig. 3 | Shapley values (SHAP) values computed for the standard deviation in wind vertical velocity ($\sigma_W$)[42].** These values represent the effect of each input on the overall deviation in $\sigma_W$ from its base value. **a–c** Derived sampling the whole globe at different pressure ($p$) levels, whereas in panel (**d**) sampling is restricted to latitude between 60N and 90N. Each dot is a instance of the sampled values. All inputs were standardized as described in ref. 21. Red dots to the positive side of the SHAP axis indicate that increasing the input causes a positive deviation from the average (vice versa for blue dots). Inputs to the Wnet model are the Richardson number ($R_i$, dimensionless), total scalar diffusivity for momentum ($K_M$, in m²s⁻¹), the 3-dimensional wind velocity ($U$, $V$ and $W$, in m s⁻¹), the water vapor, liquid and ice mass mixing ratios ($Q_v$, $Q_{liq}$ and $Q_{ice}$ in Kg Kg⁻¹), air density ($\rho_a$, in Kg m⁻³), and air temperature ($T$, in degrees K). The subscript "sfc" denotes surface values at each location.

vertical resolution. An alternative input set, which replaces scalar diffusivity and the Richardson number with PBL height (PBLH), was developed to accommodate a wider range of meteorological datasets. The core of Wnet remains unchanged, with only a transformation layer added to map the alternative inputs. Here, we focus on the original inputs, while Fig. S8 shows the Shapley values for the alternative set.

Figure 3 displays the Shapley values for each input of the Wnet model. While all input variables significantly influence $\sigma_W$, water vapor mixing ratio ($Q_v$), temperature ($T$), and density (both at the grid level and the surface) consistently show the highest Shapley values. These effects suggest the presence of active convection amplifying variability, as humidity at the surface and mid-levels strongly influences the formation and evolution of convective storms[43]. Increased humidity also causes fluctuations in the virtual potential temperature, which dictates the buoyancy of air parcels[5]. As satellite retrievals and reanalysis products indicate a positive trend in $Q_v$ near the surface over the last four decades[41], the connection between water vapor and $\sigma_W$ explains some of the positive trends depicted in Fig. 1. These are likely related to a rise in sea surface temperature (SST) of about 0. 1°C per decade in the Tropical regions of the low and middle troposphere, although they are also influenced by changes in atmospheric winds, the hydrological cycle, aerosols, and volcanic activity[41,43].

The Shapley values for other input variables reflect the physical processes driving variability at different vertical levels. Near the surface (900 hPa), the presence of liquid water significantly impacts $\sigma_W$

highlighting the strong connection between cloud formation, mixing, and turbulence[44]. Positive trends in $\sigma_W$ in persistent stratocumulus regions in the northern hemisphere may be linked to increased cloudiness from enhanced surface evaporation, although such is not the case for cloudy regions along the shores of South America and South Africa. At 500 hPa, vertical winds and the presence of ice condensate become increasingly important. These factors are associated with the presence of tropical middle and high clouds, which explain the trends depicted in Fig. 1, as reanalyses typically show positive trends in high cloud in this region[28]. This pattern is largely reproduced at 250 hPa, although some tropical regions exhibit negative trends, possibly due to unusually high cloud amounts during major El Niño events in 1982–1983 and 1997–1998[28].

Enhanced convection, however, cannot account for the negative trends in $\sigma_W$ observed in polar regions. Instead, the Shapley values found at 500 hPa in the Arctic suggest that horizontal winds may play a more significant role in variability than surface-initiated convection. It is likely that the negative trend in $\sigma_W$ in the polar regions is linked to the weakening of the polar jet stream, a phenomenon associated with Arctic amplification[45]. This connection between large-scale dynamics and microscale variability is known as chaotic advection, resulting from exponentially diverging paths of tracers within a grid cell[43,46]. Thus, the evolving meteorological patterns, like jet-activity, have systematically impacted the intensity and frequency of small-scale processes, such as turbulence and cloud formation.

## Discussion

The spatial pattern shown in Fig. 1, with positive trends in $\sigma_W$ across the tropics and negative trends in the polar regions, suggests that large-scale meteorological conditions influence small-scale processes such as mixing, cloud formation, and boundary layer dynamics. These findings are consistent with reports of increasing clear-air turbulence in recent decades likely resulting from increasing temperatures[47,48]. Cloud formation is one process where this connection could have significant climatic implications, as changes in $\sigma_W$ driven by long-term shifts in water vapor and temperature can systematically affect cloud droplet and ice crystal formation rates[1], thereby altering hydrometeor size, cloud cover, and precipitation patterns. While it is well established that large-scale cloud properties such as cloud fraction and liquid water path (LWP) are sensitive to changes in SST and water vapor shifts, their impact on microphysical properties like cloud droplet number and size remains largely unexplored. This represents a potential feedback mechanism mediated by the microphysical properties of clouds and small-scale dynamics.

Following methods used to quantify radiative impacts of aerosol-cloud interactions[49,50], we estimate the potential impact of the cloud microphysical feedback as the change in backscattered shortwave ($\Delta SW$) radiation at the top of the atmosphere due to variation in $\sigma_W$. We focus on cloud droplet formation in low-level clouds, where the radiative effect of $\sigma_W$ is expected to dominate[2]. Mechanistically, a higher $\sigma_W$ leads to a higher droplet number concentration ($N_d$), increasing the cloud optical depth, and the radiation reflected back to space[51]. Although ice crystal formation is also affected by $\sigma_W$[52], its radiative impact is likely limited[53] because a large fraction of cirrus clouds originate from convective detrainment[54]. We likewise neglect changes in large-scale cloud properties (LWP and cloud fraction) driven by variations in $N_d$ from $\sigma_W$, as their radiative impact is relatively weak[49]. To estimate the long-term change in $\sigma_W$, we use a third reanalysis, ERA-20C, which extends back to 1900, assimilates surface pressure and marine winds, and shares much of its modeling framework with ERA5[55].

Figure 4 shows that over roughly the past century, $\sigma_W$ has increased by about $0.052 \pm 0.014 \text{ms}^{-1}$, corresponding to a global mean top-of-the-atmosphere radiative forcing of approximately $-0.10 \pm 0.04 \text{W m}^{-2}$. The effect is larger when using MERRA-2 data ($\Delta SW = -0.13 \text{W m}^{-2}$, Fig. S9) compared to ERA5 ($\Delta SW = -0.07 \text{W m}^{-2}$, Fig. S10), which is expected given the weaker trends in $\sigma_W$ in ERA5 (Section "Results"). Uncertainty in $\Delta SW$ is estimated from a sensitivity analysis varying the vertical level for $\sigma_W$, present day and beginning of the century years, and the reanalysis dataset (see Table S3).

$\Delta\sigma_W$ between 1900 and 2020 is dominated by positive shifts over the oceans, particularly in tropical regions and along the storm tracks of the southern hemisphere (SH). However, regions of persistent stratocumulus in the SH (off the Namibian Desert and the west coast of South America) show a reduction in $\sigma_W$. The extratropics also display

slight negative changes in $\sigma_W$, particularly over land. These patterns resemble those in Fig. 1, though somewhat amplified. The ERA5-derived dataset (Fig. S10) also shows slight negative $\Delta\sigma_W$ in the stratocumulus regions of the northern hemisphere, especially off the west coast of North America.

Besides the relative change in $\sigma_W$, $\Delta SW$ is also controlled by the susceptibility of $N_d$ to $\sigma_W$ ($\beta = \frac{\partial \ln N_d}{\partial \ln \sigma_W}$), estimated over the period 1980–2022 by allowing $\sigma_W$ to vary each year, while calculating $N_d$ using annually repeating mean climatological aerosol fields from MERRA-2, as described in the "Methods" section[50]. This approach isolates the effect of $\sigma_W$ from changes in $N_d$ due to evolving aerosol emissions. The resulting spatial distribution of $\beta$ is shown in Fig. 4. Low $\beta$ values indicate an aerosol-limited regime and are associated with high $\sigma_W$ (Fig. 1), whereas high $\beta$ implies that droplet formation is more strongly controlled by $\sigma_W$. As a result, relatively pristine regions of the tropics and the SH, despite exhibiting high $\Delta\sigma_W$, show a limited radiative effect. Instead, $\Delta SW$ is more negative in regions of high susceptibility, such as the subtropics and industrial centers in South and Central Asia. Conversely, the negative $\Delta\sigma_W$ observed offshore of South Africa and South America produces a relatively large positive $\Delta SW$.

The positive $\Delta SW$ observed offshore of the SH continental regions warrants attention, as it exemplifies the mechanism proposed here. This pattern is seen when using both MERRA-2 and ERA5 (Figs. S9 and S10). Calculating $\Delta SW$ for different decades between 1980 and 2020 (Fig. S11) and using different years from the ERA-20C data (Fig. S12) results in similar patterns, indicating that this signal does not stem from multiyear variability or climate oscillations such as El Niño events. To investigate the drivers of such a pattern, we analyze the Wnet input variables using the alternative set that includes PBLH. Figures S13 and S14 indicate a general decrease in PBLH and grid-scale $W$ across much of the subtropics over the past century. In particular, the spatial distribution of $\Delta$PBLH resembles the $\Delta\sigma_W$ pattern of Fig. 4. An explanation for this is that a shallow PBL is associated with enhanced stability hence reduced vertical velocity variability and $\sigma_W$. The causes of PBL shallowing are complex and may relate to a strengthening of the Hadley cell[56], but a full investigation is beyond the scope of this work. These results however, highlight how large-scale climate changes influence $\sigma_W$ and how the resulting cloud microphysical feedback affects climate predictions. For instance, circulation changes that reduce PBLH may lower $\sigma_W$ and cause a thinning of clouds, which would increase local temperatures. This would then tend to deepen the PBL, hence introducing a negative feedback on climate. Conversely, if the PBL deepens due to increased SSTs, both $\sigma_W$ and cloud optical depth may increase, which would tend to cool the surface. Importantly, changes in PBLH can also affect cloud fraction and thickness through non-microphysical processes, which may either offset or amplify the microphysical component of the cloud feedback. This has implications for marine cloud brightening geoengineering

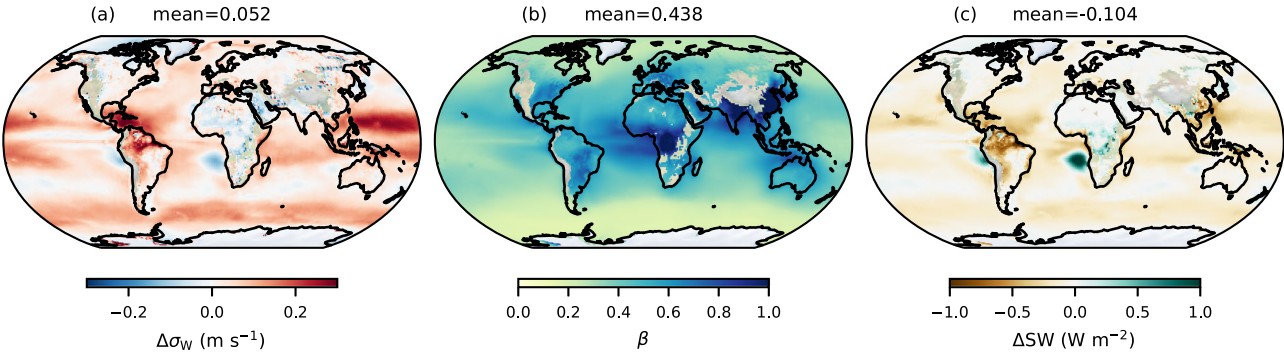

**Fig. 4 | Long-term change in the standard deviation in the vertical wind velocity ($\sigma_W$) and its impact on radiative forcing.** Shown are **a** the change in $\sigma_W$ over the period 1900–2020 ($\Delta\sigma_W$), **b** the droplet number concentration ($N_d$) susceptibility to $\sigma_W$ ($\beta = \frac{\partial \ln N_d}{\partial \ln \sigma_W}$), and **c** the associated shortwave radiative forcing ($\Delta SW$).

projects[57], where shifts in $\sigma_W$ could either counteract or amplify the intended cloud-seeding effects.

The representation of subgrid-scale variability in vertical wind velocity in atmospheric models plays a critical role defining cloud formation rates, turbulent transport and boundary layer mixing. Here, we present extensive datasets offering a detailed, global description of $\sigma_W$ from 1980 to 2022 at a high temporal resolution. The data were created using a deep learning model combining high-resolution simulations with long-term observations of $\sigma_W$, applied to the MERRA-2 and ERA5 reanalyses. We rigorously validated our approach using independent in situ data, collected using different measurement techniques, and for a wide variety of cloud regimes and atmospheric environments. These new datasets merge high-resolution modeling, reanalysis data, and observations from various techniques, providing an observations-constrained global estimate of $\sigma_W$. These estimates have the potential to evaluate turbulent transport models, drive parameterizations of aerosol activation in GCMs and chemical transport models, and complement estimates of turbulence indices for air traffic safety.

Analysis of the time evolution of $\sigma_W$ revealed significant trends over the past four decades, with a net increase over the Tropics and the Midlatitudes and a decrease in the Arctic and off shore of continental regions in the southern hemisphere. Explainable machine learning indicated that these trends are driven by large-scale changes in the global distribution of temperature, water vapor and winds. This strongly suggests that long-term shifts in large-scale meteorological patterns drive changes in small-scale variability. We demonstrate that this could lead to trends in cloud microphysical properties through the modification of hydrometeor formation rates, exerting a substantial influence on climate. Modified cloud microphysical properties in turn alter the radiative balance of the Earth. This new microphysical feedback mechanism should be considered in climate assessments and geoengineering projects.

Although the estimate of $\sigma_W$ presented here is grounded in observational data, it is sensitive to the global distribution of atmospheric variables, particularly the vertical profile of water vapor. There are still discrepancies on the estimates of water vapor amount among different retrievals and reanalyses, introducing uncertainty into the estimates of the absolute trend in $\sigma_W$. Proper scaling must be applied when comparing our generated dataset to horizontal resolutions other than the ~0.5°, inherent in the Wnet model. While the training algorithm used in the Wnet model filters experimental error, there is still some residual error in the approximations made in its development, for example in the collocating of observational data onto a GCM grid. This is estimated to be ~15%[21] though it may be larger in convective regimes. These challenges could be addressed in future research by expanding the observational and modeling data during training, particularly enhancing the representation of deep convective clouds. Long-term modeling would also help elucidate the main drivers of the microphysical feedback and the potential role of $\sigma_W$ on the evolution of ice microphysical properties. This study integrates datasets from multiple sources to improve understanding of a key variable in atmospheric modeling that remains poorly constrained. Our findings demonstrate a clear two-way connection between large- and small-scale processes in shaping climate evolution. Incorporating such linkages in atmospheric models may help reduce uncertainty in climate assessments.

## Methods

### Estimation of $\sigma_W$

A schematic of the workflow used to produce the global datasets of $\sigma_W$ is shown in Fig. S2. The Wnet model was developed in two stages. First, the neural network, termed Wnet, was trained using $W$ output from a global storm-resolving simulation ($\Delta x$ ~ 6 km), with $\sigma_W$ computed over areas corresponding to ~50 km grid cells. This initial model serves as a

feature extractor, identifying meaningful relative patterns in $\sigma_W$. However, due to the relatively coarse resolution of the GSRM, it tends to underpredict the magnitude of $\sigma_W$. To correct for this, a refinement step was introduced, where Wnet was further trained using long-term observations, which derive $\sigma_W$ from high-frequency radar and lidar retrievals of $W$, covering most atmospheric motion relevant to cloud formation. The $W$ fields were processed using the average wind at each site to represent $\sigma_W$ over a ~50 km grid cell, as detailed in ref. 21. Different sites were selected from around the world to sample a diversity of cloud and metereological regimes as listed in Table S2 and depicted in Fig. S1.

The refinement against observations followed a generative approach, incorporating a second neural network, called the discriminator, which guides the training process[21,22]. During this stage, Wnet does not learn directly from the observational data but instead from the representation of $\sigma_W$ learned by the discriminator. This architecture allows the discriminator to filter out experimental error, exposing Wnet only to meaningful, structured relationships in the data. As a result, the model filters out noise and develops a robust parameterization of $\sigma_W$.

Once trained, we carried out the inference step. Using MERRA-2 to inform the input to Wnet, we computed $\sigma_W$ using instantaneous values at 3-hourly intervals, on 72 model levels, then interpolated to 44 pressure levels. The input to the Wnet model (denoted as $X$ in Fig. S2) consist of a 14-dimensional vector including the Richardson number ($R_i$, dimensionless), total scalar diffusivity for momentum ($K_M$, in m²s⁻¹), the 3-dimensional wind velocity ($U$, $V$ and $W$, in m s⁻¹), the water vapor, liquid and ice mass mixing ratios ($Q_v$, $Q_{liq}$ and $Q_{ice}$ in Kg Kg⁻¹), air density ($\rho_a$, in Kg m⁻³), and air temperature ($T$, in degrees K) at the resolution of the MERRA-2 reanalysis (0.5° × 0.625°). Surface $Q_v$, air density, $K_m$ and $R_i$ were concatenated to each level to account for the effect of orography and convection on upper tropospheric $W$ variability. Deseasonalized anomalies were calculated relative to the 1980−2022 long-term monthly climatology. Linear trends were obtained by performing regression on monthly means at each grid cell.

A second global dataset was generated for the period 1980−2022 using output from the ERA5 reanalysis[26] to drive Wnet. The instantaneous input state was extracted at 3-hourly intervals with a nominal resolution of 0.25° and 37 pressure levels. Unlike MERRA-2, the ERA5 output does not include the parameters $R_i$ and $K_m$. To address this, Wnet was adapted as a foundational model for subgrid-scale variability by introducing a transformation dense layer before Wnet to translate the input state, as shown in Fig. S3. This new layer consisted of 128 nodes, utilized Leaky ReLU activation, and was trained on MERRA-2 data while keeping the original Wnet layers frozen. The input vector for ERA5 included 11 variables: PBL height ($PBLH$, in m), total integrated water vapor ($TPW$, in kg m⁻²), 2-m temperature ($T2M$, in K), 3-dimensional wind velocity components ($U$, $V$, $W$, in m s⁻¹), water vapor, liquid, and ice mass mixing ratios ($Q_v$, $Q_{liq}$, $Q_{ice}$, in kg kg⁻¹), air density ($\rho_a$, in kg m⁻³), and air temperature ($T$, in K). Two-dimensional variables ($PBLH$, $TPW$, $T2M$) were broadcasted to three dimensions. After training, the ERA5 input was used to estimate $\sigma_W$. A third, smaller $\sigma_W$ dataset was derived from the ERA-20C reanalysis[55], spanning the years 1900−1905, computed using instantaneous values at 6-hourly intervals, at a resolution of 1° on 37 pressure levels.

### Validation against observational data

Two sources of data were used to validate the $\sigma_W$ datasets, direct measurements taken during air flight campaigns and ground-based remote sensing retrievals at specific locations around the world. Table S1 describes the field campaign data. Measurements of $W$ were used to estimate $\sigma_W$ by computing the standard deviation over a time window of about 430 s; roughly the time it takes for the aircraft to

cover an area equivalent to a standard GCM grid cell (i.e., 50–100 km). $W$ was measured at a frequency of 1 Hz, corresponding to a spatial scale $\Delta x$ ~ 300 m, with an estimated error of about 0.1 to 0.3 m s⁻¹. The measurement techniques and the quality screening of the data are detailed in ref. 13. The $\sigma_W$ datasets derived from MERRA-2 and ERA5 were interpolated to the spatial location and time of each campaign flight, generating one-dimensional curtains along the aircraft trajectories. Due to the discrepancy between the gridded data resolution and the frequency of measurements, it was challenging to pinpoint exactly the altitude of the aircraft at each time. Instead, $\sigma_W$ statistics were computed for each field campaign by extracting $\sigma_W$ for each curtain within two standard deviations of the measured mean temperature for each flight, and for pressure ranging between 50 and 600 hPa thus ensuring that only data for steady flight conditions were utilized for validation.

The remote sensing data used for validation is summarized in Table S2, with screening criteria detailed in ref. 21. The dataset includes lidar and radar retrievals of $W$ with a frequency between 1 and 10 Hz, and a corresponding spatial resolution $\Delta x$ ~ 30–300 m, covering periods between 6 months and 10 years, depending on the site. The mean horizontal wind at each site was used to estimate $\sigma_W$, typically calculated over 5 min −20 min intervals. Radar retrievals focused on cloudy cirrus layers, while lidar observations were limited to altitudes below 4 km. A single deep convective site (MAO) utilized scanning radar to profile $W$ from the surface to 14 km. The estimated error in $W$ is -0.1 to 0.2 m s⁻¹ [11,12,37]. The MERRA-2 and ERA5-derived $\sigma_W$ datasets were interpolated to match the location, time and altitude of each measurement and compared with observations. Overall, the observed $\sigma_W$ distributions were well represented in the datasets, with discrepancies within the experimental error. It is important to note that, unlike the aircraft data, the remote sensing retrievals were used in the development of Wnet, albeit through a different method.

### Radiative forcing estimation

Since changes in $\sigma_W$ mainly impact $N_d$, we adapted a methodology commonly used in the study aerosol-cloud interactions [49,50] to calculate the radiative forcing associated with changes in $\sigma_W$. This is given by:

$$\Delta SW = -SW_{\text{down}} \times C_f \times A_c \times (1 - A_c) \times \frac{1}{3} \frac{\partial \ln N_d}{\partial \ln \sigma_W} \times \Delta \ln \sigma_W \quad (1)$$

where $SW_{\text{down}}$ is the downwelling shortwave flux at the top of the cloud layer (assumed here to be equal to $SW$ at the surface[49]), $C_f$ is the liquid cloud fraction, and $A_c$ is the cloud albedo. Equation (1) assumes that the primary radiative impact of $\sigma_W$ arises from changes in cloud albedo, while adjustments in LWP and cloud fraction can be neglected. This assumption is justified because changes in $N_d$ induced by $\sigma_W$ are relatively small compared to other factors like aerosol emissions, and LWP is only weakly sensitive to $N_d$[49]. Climatological values averaged over 2015–2020 were used for $C_f$, $SW_{\text{down}}$, and $A_c$. $C_f$ and $SW_{\text{down}}$ were obtained from the MCD06COSP_M3_MODIS[58] and CERES EBAF-TOA[59] datasets, respectively. MERRA-2 data were used to estimate $A_c$, as it has been found to agree well with satellite observations[60]. $\Delta \ln \sigma_W$ was estimated as the difference between the global mean $\sigma_W$ averaged over 2015–2020 from the combination of the MERRA-2 and ERA5 climatologies, and over 1900–1905 from the ERA-20C reanalysis[55]. Uncertainty in $\Delta SW$ was estimated by varying the vertical level for $\sigma_W$ and $\beta$ (875–975 hPa), present day (2015–2020) and beginning of the century (1900–1905) years, and the reanalysis dataset for a total of 378 combinations (see Table S3). This resulted in a calculated range of $\Delta SW = [-0.212, -0.043]$ W m⁻² with a standard deviation of 0.037 W m⁻² ($\Delta \sigma_W = [0.039, 0.093]$ m s⁻¹ with a standard deviation of 0.014 m s⁻¹).

To calculate the susceptibility of $N_d$ to $\sigma_W$, defined as $\beta = \frac{\partial \ln N_d}{\partial \ln \sigma_W}$, we followed the methodology detailed in ref. 50. This approach uses an auxiliary neural network, MAMnet[61], to predict the aerosol size distribution (ASD) from MERRA-2 assimilated aerosol products. MAMnet emulates the Modal Aerosol Module [MAM7[62]] and predicts number concentration and composition for seven internally mixed lognormal modes (accumulation, Aitken, coarse/fine dust, coarse/fine sea salt, and primary carbon matter). It was trained on 5 years of data from a simulation of the Goddard Earth Observing System (GEOS) model[61] implementing MAM7 and validated against ground-based observations. To isolate the effect of $\sigma_W$, a fixed, monthly mean climatological ASD was developed using the entire MERRA-2 dataset. This ASD was annually repeated, and, together with $\sigma_W$ from Wnet, used to predict $N_d$ for 1980–2022 following the approach detailed in ref. 63. The parameter $\beta$ was then obtained as the slope of a linear regression of $\ln N_d$ against $\ln \sigma_W$ across the time dimension, on monthly means, at each grid cell. Since both $\sigma_W$ and $\beta$ are three-dimensional fields, $\Delta SW$ is reported as the mean over the five lowest model levels (875–975 hPa). All variables were regridded to 1° resolution to match the ERA-20C grid.

## Data availability

All data generated in this work is publicly available through https://portal.nccs.nasa.gov/datashare/Wstd-MERRA2/Wnet. Field campaign data can be found at https://data.mendeley.com/datasets/pr28vks52k/1[64]. Datasets to reproduce the figures presented in this work can be found at https://zenodo.org/records/17676312. The MERRA-2 Reanalysis is publicly available from https://disc.gsfc.nasa.gov/. The ERA5 dataset was downloaded from https://cds.climate.copernicus.eu. The ERA-20C was downloaded from https://climatedataguide.ucar.edu/climate-data/era-20c-ecmwfs-atmospheric-reanalysis-20th-century-and-comparisons-noaas-20cr. MODIS data was obtained from https://ladsweb.modaps.eosdis.nasa.gov/missions-and-measurements/products/MCD06COSP_M3_MODIS/. CERES data was obtained from https://asdc.larc.nasa.gov/project/CERES/CERES_EBAF-TOA_Edition4.1.

## Code availability

The GEOS-5 source code is available under the NASA Open Source Agreement at https://github.com/GEOS-ESM. The Wnet model is available at https://github.com/dbarahon/Wnet[65].

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

## Acknowledgements

Resources supporting this work were provided by the NASA High-End Computing (HEC) Program through the NASA Center for Climate Simulation (NCCS) at Goddard Space Flight Center. Keras and Tensorflow libraries were obtained from https://keras.io/. Maps were created using the NCAR Command Language (Version 6.6.2) Software. (2019). Boulder, Colorado: UCAR/NCAR/CISL/TDD. https://doi.org/10.5065/D6WD3XH5. The SHAP python package was used to conduct the explainable machine learning analysis as described in https://shap-lrjball.readthedocs.io/en/latest/index.html. Donifan Barahona was funded by the NASA MAP (grant: NNH20ZDA001N-MAP) and the NASA ACCDAM (WBS: 281945.02.31.04.39) programs. Katherine Breen was funded by the NASA MAP program, grant NNH20ZDA001N-MAP. Minghui Diao and Derek Ngo were funded by NASA ACCDAM 80NSSC21K1457 and by NASA MOSAICS 80NSSC24K1616. Minghui Diao, Flor Vanessa Maciel, and Ryan Patnaude were funded by NSF AGS 1642291 and NSF OPP 1744965.

## Author contributions

D.B. conceived, directed, and developed the work. K.B. co-developed the Wnet model. D.N. and M.D. developed the curtains along the aircraft trajectories. M.D., V.M., and R.P. compiled field campaign data and screened it for quality.

## Competing interests

The authors declare no competing interests.
