## [Transparent Peer Review file · Nature Communications]

Trends in vertical wind velocity variability reveal cloud microphysical feedback

Corresponding Author: Dr Donifan Barahona

Version 0:

Reviewer comments:

Reviewer #1

(Remarks to the Author)

I have three primary criticisms of this article. After these are addressed, I think it will be suitable for publication. I have entered major revisions.

Comment 1:

There needs to be more information on how σ_w actually influences a global model. Presumably this is via the parameterization of the turbulent flux terms? Is σ_w diagnosed in the cumulus parameterization, and in your sensitivity experiments you are simply replacing the diagnosed value with your PI and PD values?

Comment 2:

Fig 2 and the associated discussion would benefit from a correlation analysis. Are the differences in σ_w among the field projects/locations statistically significantly correlated with the analogous differences computed from MERRA-2?

Comment 3:

The B24 model was derived using a 5 km grid spacing model, which under-resolve a lot of convective motions including those in tropical deep convection, shallow convection, and smaller scale cloud features. What are the implications of this limitation? At least some discussion of this issue is warranted.

Reviewer #2

(Remarks to the Author)

This study relates observed variability in vertical air motion (σ_w _obs) to model-parameterized sub-grid-scale vertical air motion (σ_w _model) using a deep learning technique. It then uses the global, multi-decade model (the MERRA-2 reanalysis) to predict σ_w globally and over multiple decades. A multi-decade (1980-2023) global trend in σ_w is described. This trend is attributed to changes in the climate system.

This trend is intriguing. A Shapley Additive explanations (SHAP) analysis shows that the trend is related to global warming and coincident increases in water vapor mixing ratio. Assuming that the described trend in σ_w is accurate, one wonders whether this finding is noteworthy. Climate system feedbacks, in particular cloud feedbacks, do involve σ_w , e.g. mid-tropospheric σ_w is largely due to deep convection, so an independent measurement of a σ_w trend would be most noteworthy, as it would explain the observed global increase in precipitation rate (and the rate of hydrological cycling). But this paper does not present an independent observational finding. Rather, since σ_w is parameterized in MERRA-2, the documented trend simply is a dynamically consistent consequence of an increase in global precipitation rate captured by MERRA-2 and other reanalyses. So, I argue that the key finding of this paper lacks novelty. It merely confirms an already established positive trend in convective precipitation.

This paper, and references cited, generally provide the details necessary for the method to be followed work and the results to be reproduced. However questions do remain, e.g. what assumptions are made to reduce Doppler hydrometeor vertical velocity (from radar) to vertical air motion. The key reference [20], on which this study builds, does not detail these assumptions.

While the paper is well written, the study in-depth, and the findings original, I believe there are flaws in the data analysis, interpretation and conclusions. First, the training dataset of observations have unacknowledged uncertainties or limitations, adding noise to σ_w _obs. Profiling Doppler Lidar (DL) observations are generally limited to the boundary layer (BL) where σ_w is fundamentally different from that in the free troposphere, where the deep learning technique is mostly applied. Doppler radar (DR) data are used as well, but unknown variations in hydrometeor fallspeed contribute significantly to the observed variability, hence the DR variability is a poor proxy for σ_w . Also, DR data are limited to precipitating regions only, while DL are limited to clear air, so their σ_w values represent different atmospheric conditions. Independent airborne estimates of σ_w are used to validate the MERRA-2-based (predicted) σ_w _model, but the airborne measurements are made elsewhere (above the boundary layer, outside precipitating areas, in clouds that neither DL nor DR can sample, and mostly in the upper troposphere, as indicated by the field campaigns listed in Table 1), and with an unacknowledged uncertainty (especially on the low-frequency side). σ_w values in the upper troposphere typically are much lower than in the BL, so the validation does not capture the spectrum of variability. Second, the frequency spectrum of data from ground-based profiling systems (DR and DL) and that from airborne gust probes differ, and is different from models' sub-grid-scale vertical velocity. Third, I question the robustness of the key finding, that in the last four decades, the global troposphere has experienced an increase in σ_w of about 1% per decade, reaching up to 10% per decade in some regions. This is according to the MERRA-2 dataset, subjected to a ML technique trained on data that have measurement uncertainty and lack representativeness. The impact of measurements' uncertainties and excessive clustering in the physical and the spectral domains is not analyzed. Fourth, MERRA-2 is one reanalysis. The same technique was used on one other reanalysis, ERA5, and the results are quite different (Fig. A3): the trend essentially vanishes above the boundary layer (500 and 250 hPa) in ERA5, and the spatial patterns across the globe are different (Fig. 1 vs Fig. A3). The differences in values between ERA5 (monthly-mean data) and MERRA-2 (6 hourly) makes sense: monthly-mean data do not capture the synoptic variability in σ_w . But that does not explain the differences in trend. Given this disagreement, I would recommend the use of full-time-resolution ERA5 data, and other global or regional reanalyses, to build further evidence.

Some of the terminology used in the paper is unfortunate and inconsistent with expected standards. For instance, vertical "wind" is used, where the term wind refers to horizontal air motion. I acknowledge that other papers have used the expression "global cloud resolving models", but of course, global models cannot truly resolve cloud processes: the latest DNS models, which capture cloud microphysical processes, have meter-scale volumes at most. I think they refer to global convection-permitting models, which exist, e.g. the E3SM SCREAM. But then, MERRA-2 is not one of them. Maybe they simply refer to numerical models in which cloud processes are parameterized. That has been done in climate models since the beginning, going back to the early CMIPs. Statements like "cloud formation occurs at scales around 100 m" depart from the established core knowledge, in this case understanding of the multiscale nature of clouds. There is no "Pacific Cold Tongue at 250 hPa". There are many other examples.

In summary, I believe that these weaknesses and flaws prohibit publication.

Version 1:

Reviewer comments:

Reviewer #3

(Remarks to the Author)

Review of the manuscript „Trends in vertical wind velocity variability reveal cloud microphysical feedback“ by Barahona et al.

In the manuscript „Trends in vertical wind velocity variability reveal cloud microphysical feedback“ Barahona et al. apply their recently published generative AI model (Wnet) to predict the spatial variability (standard deviation) of vertical wind σ_w from large scale meteorological conditions from reanalysis (MERRA-2 and ERA-5). They find significant increases in σ_w that are particularly strong over low (900hPa) and mid-level (500hPa) oceanic regions. This finding suggests increasing variability in supersaturation and therefore changes in aerosol activation and cloud microphysics, and implies a (comparably small) negative feedback. This finding is important and justifies the publication in Nature Communications.

My main task as a reviewer was to evaluate the authors response to the criticism by reviewer 2 (R2), who, in her/his review was very skeptical about the methodological approach, i.e. about the ability of the Wnet to accurately predict σ_w . After reading the manuscript, the reviews and the authors responses, I am convinced that the criticism of R2 is mainly founded on an incomplete understanding of the details of the generative AI method developed by the authors. In my opinion, the authors have done substantial work to improve the communication and clarity of the manuscript in this regard.

I do question the authors focus on one reanalysis to predict σ_w , though. In the manuscript, the results of σ_w data generated with the large scale fields from MERRA-2 are highlighted, I guess because the trend signal is much more pronounced than in ERA-5. In my opinion this is not adequate and seems a bit like cherry picking of the results to me unless there is a physical reason why we should trust MERRA-2 more than ERA5. If that is not the case I believe the two results should be equally weighted in the interpretation and in the headline statements. In this aspect, I agree with R2, comment 4, and think it is not appropriate to overrepresent the findings from the reanalysis that produces the stronger trend signal in σ_w . I acknowledge that ERA5 is now discussed more in the manuscript than initially, but still there seems to be a focus on the aspects where the two data sets agree rather than clearly state the differences between them.

(Remarks on code availability)

Reviewer #4

(Remarks to the Author)

Review of Barahona et al. - Trends in vertical wind velocity variability reveal cloud microphysical feedback.

I'm referring to the revised manuscript here.

Responses to the comments of the other reviewers.

Reviewer#1

There was some confusion (comment#1) about what the ERF that was calculated represented, but I think that the main manuscript has been changed to make it clear now that the ERF value is just due to the microphysical effect on droplet/ice activation. However, this is still not the case in the abstract and I have made some suggestions for that below.

I think that comment#2 has been addressed reasonably.

Comment#3 concerns the fact that the 5km resolution of the high resolution model won't resolve a lot of the relevant motions and suggests more discussion. I agree with this point. Some discussion has been added, but I think that a little more discussion would be helpful. I made some similar comments below for Section 2., which are repeated here:-

- Here it would be good to discuss that there is a large range of scales of updrafts/turbulence (from global circulations, though to deep convection, shallow convection, stratocumulus, small-scale mixing and turbulence). And how sub-grid σ_w will depend on the resolution being considered – i.e., what is the size of the "grid". In this study you use a 5km model, but also include some smaller scales with the observations – what resolutions/scales do the observations used to build the model cover? And what is the resolution of the aircraft observations used to test the model? The results for the ERF calculation rest on changes in stratocumulus clouds in the sub-tropics, off the west coasts of S America, N America and SW Africa. However, there is little discussion about the spatial scales of vertical winds in these regimes and about how well your model captures them. One approach used in the past, which should be discussed in this paper, has been to use very high resolution (~35-150m) LES models to quantify σ_w at stratocumulus-relevant scales (<https://agupubs.onlinelibrary.wiley.com/doi/full/10.1002/2013JD021218>). Do the observations you used to build and test the model cover these smaller scales for stratocumulus clouds?

Reviewer#2

Some of the comments may have come from unclear explanations in the introduction and methods sections and there has been some effort to improve that. However, this aspect could still be improved further and I've tried to make some suggestions for that below. It is still not clearly stated whether the sub-grid parameterized σ_w of the high resolution model is used along with the resolved updrafts (I don't think it is, but I'm not sure that it's stated clearly).

The comment "While the paper is well written, the study in-depth, and the findings original, I believe there are flaws in the data analysis, interpretation and conclusions.", also questions the types of regime sampled by the remote sensing used to build Wnet and the aircraft obs used to test it. I agree with this aspect – it's not clear which types of cloud/meteorological regimes have been sampled by the aircraft observations from the tables and text provided. There is some useful information in Table S1 for the ground station data, but more information on the cloud types sampled would be useful – does "PBL" cover stratocumulus and trade cumulus, or just clear-air observations? Nor is it clear what the resolution of the observations was. Some more discussion on this and the limitations is warranted (linking to the comments above about stratocumulus). I think the other comments have been addressed well.

My own comments

One of the key results (the "ERF" forcing due to σ_w changes) rests on the difference in the σ_w climatologies in the PD run compared to that from the coupled PI run being representative and not just a random chance result. E.g., if different 10 year samples were chosen or if longer samples were used, would the results be the same? I don't think that 10 years is long enough to capture the variability that is possible in a coupled simulation such as the PI simulation. In a coupled run the SST patterns, sea-ice amount, etc. will vary a lot and 10 years won't capture this. Can you use the full 180 years? Even for the prescribed SST runs it is too short since there will be a lot of natural variability in the climate that you would need to sample over. The exact 10-year period chosen may also influence the mean climatological winds, circulation, etc. and it's possible that different 10-year periods might give different results. I would expect a longer period to be used (around 30 years). It would be good to compare results using 10, 20, 30 years to make sure they are robust. And similarly in the coupled PI run, although you will probably have to extend to even longer periods for that.

Possibly related to the comment above is the concern that the differences in σ_w between the PD and PI runs don't seem consistent with those between the +4K and PD runs (Fig. S7). There is also little resemblance to the long-term trends. This is a concern as it suggests that the impact of warming and a changing climate is not being captured consistently with the model experiments. There are some more detailed comments on this below, but the longer experiments above should help to address the issue. More discussion on these discrepancies in the results is needed.

Detailed comments

Abstract

"This effect is however difficult to represent"

- Better without the "however" here since it is also used later in the sentence.

"however can be addressed by machine learning." -> however, it can be addressed...

“indicating enhanced turbulent atmosphere” -> indicated enhanced atmospheric turbulence?

“suggesting a feedback connection between enhanced warming, turbulence and cloud microphysics, with an associated radiative forcing of about $-0.1 \pm 0.21 \text{ W m}^{-2}$, slightly mitigating greenhouse warming.”

- You haven't introduced the effect on microphysics here yet and it's also not clear here that the “forcing” is solely due to the microphysical effect. I suggest something like this – “suggesting a feedback connection between enhanced warming and turbulence, which in turn has a microphysical effect through the activation of more cloud hydrometeors. The forcing associated with the microphysical effect is estimated here to be about $0.1 \pm 0.21 \text{ W m}^{-2}$, slightly mitigating greenhouse warming.”

Introduction

“modern atmospheric models”

– You should specify the subset of models being talked about here rather than later in the paragraph - this only applies generally to global models. E.g., LES models have much higher resolution.

“Nevertheless, the GSRM simulations and the extensive data they produce represent a wealth of information that can be used to understand and model σ_w . Artificial intelligence techniques and data-driven algorithms like deep learning, are well-suited for this task. Unlike traditional numerical simulations or theories, these algorithms learn from data to make predictions.”

– You haven't introduced observational data yet – surely this is necessary to allow the AI approach to improve the predictions of the GSRM simulations?

“it was trained on results from global storm-resolving model simulations,”

- It would be helpful to quote the resolution of the model here to make it clear that it is built on high-resolution data. I don't think that it is mentioned in the whole paper. Also, to avoid confusion later, it would be good to clarify in this sentence that it is trained to predict σ_w as output from the high-resolution model as a function of the low resolution meteorology (was this from MERRA or from the coarsened fields of the high-res model?)

“spatial distributino” typo.

“By integrate physical relationships” -> By integrating...

“Since the Wnet was built using data from reanalyses like MERRA-2,”

- Was it built from MERRA2 or from coarsened fields from the high-res model? If the latter then it would perhaps be better to say that the Wnet model uses input variables that can provided by MERRA2.

“Our analysis revealed”

- Better to say “We will present an analysis that shows” or similar since this sounds like you are referring to your past study.

Results

“We used the Wnet model to create three-dimensional global distributions of σ_w every 3 hours from 1980 to 2023.”

- it would be good to say that you used the meteorological fields from the 1980-2023 MERRA2 reanalysis as inputs into Wnet to create the climatology. Better here than in the next sentence.

Fig. 1 caption

- “Stippling highlights statistical significance the 95% level” -> “... at the 95% level.”

Section 2.1

- Here it would be good to discuss that there is a large range of scales of updrafts/turbulence (from global circulations, though to deep convection, shallow convection, stratocumulus, small-scale mixing and turbulence). And how sub-grid σ_w will depend on the resolution being considered – i.e., what is the size of the “grid”. In this study you use a 5km model, but also include some smaller scales with the observations – what resolutions/scales do the observations used to build the model cover? And what is the resolution of the aircraft observations used to test the model? The results for the ERF calculation rest on changes in stratocumulus clouds in the sub-tropics, off the west coasts of S America, N America and SW Africa. However, there is little discussion about the spatial scales of vertical winds in these regimes and about how well your model captures them. One approach used in the past, which should be discussed in this paper, has been to use very high resolution (~35-150m) LES models to quantify σ_w at stratocumulus-relevant scales (<https://agupubs.onlinelibrary.wiley.com/doi/full/10.1002/2013JD021218>). Do the observations you used to build and test the model cover these smaller scales for stratocumulus clouds?

Section 2.2

“depicted with in Figure 1.” – typo.

“frequency small scale processes” -typo

Section 3

“However, while turbulence indices are derived from the coarse meteorological state, σ_w indicates variability at the small scale,”

- it's not very clear what you mean here.

“corresponds to a previously unexplored feedback mechanism mediated by the microphysical properties of clouds and the small scale dynamics.”

- Although this will be inherently included in GCMs in some way since many predict TKE as a function of the grid properties, which then affects cloud droplet activation.

“We employ global atmospheric modelling”

- should say which model in the main text.

“Near the surface, the difference between the PD and PI climatologies resembles the anomalies seen in Figure 1.”

- I can't see any similarity between Fig. 1f and the lower-middle plot in Fig. S7 (900 hPa PD-PI) if that's what you're referring to. Nor for 500hPa. Plus, the PD-PI plots have the opposite sign to the SST+4K - PD plots. I would trust the latter a bit more since both are driven by SSTs and don't rely on the 10-year coupled climatologies. The signals are certainly much stronger at higher altitudes.

“Near the surface, the difference between the PD and PI climatologies resembles the anomalies seen in Figure 1.”

- Why would a decrease in cloudiness lead to less convective transport of water. Where does the cloudiness decrease?

“the radiative forcing obtained after 50 years of integration.”

- so do you use the 10 year sigma_w climatology repeated 5 times? You should make it clearer what you did.

"Enhanced σ_w in these regions leads"

- Which regions? Low-cloud stratocumulus regions? Which ones in particular? I can't really see any enhanced sigma_w in these regions in Fig. S7?

"Figure S9). In contrast, extra-tropical regions in the Pacific experience the opposite effect. Drying at lower levels weakens σ_w as convective clouds deepen in the Tropics [37]. A sensitivity run using a σ_w climatology generated from a scenario with present-day SSTs increased by 4 K [43] suggests that as drying intensifies, σ_w weakens even further."

- I can't see any evidence for this in Fig. S7? Where does sigma_w decrease in the extra-tropics in the PD experiment (vs PI). And the +4K experiments show the opposite response (vs PI) to the PD ones and stronger sigma_w in the extratropical Pacific? How do you know that the convective clouds deepen? I think that it would be best just to focus on the stratocumulus region effects rather than extratropics here? It's probably only the stratocumulus regions that are important here.

"leading to decreased coverage of stratocumulus clouds and a positive net radiative forcing on climate (see Figure S10)."

- The change in low cloud cover must come from the change in Nd since sigma_w only affects the microphysics. And so the reduction in cloud cover probably comes from the precipitation enhancement as a result of the reduced Nd? It would be good to mention this.

"could profoundly impact climate predictions."

- This seems a bit of a stretch for a 0.2 W/m² forcing with lots of uncertainty! The language needs modifying here.

Figs. 4 and S10

- It seems that the SWCE and LWCE changes should be negated to make them net incoming (instead of outgoing) TOA values for consistency with the ERF (since the value in Fig. 4 is negative giving an overall cooling)? And the definition of ERF in the caption is wrong – it should be net incoming TOA radiative forcing?

Section 5.1

"trained using long-term retrievals of"

- would be good to reiterate that these are observations.

Section 5.2

"ranging from 20 seconds to 5 minutes,"

- Is it possible to give a horizontal resolution? E.g., for a 10 m/s horizontal wind speed this would be a resolution of 200-3000m.

Section 5.3

"the MERRA-2 reanalysis and the free-running simulation."

- Make it clearer what these two are - is the reanalysis the same as the "long-term MERRA climatology run" and is the "free-running" simulation the prescribed SST PD run?

“, taken from each of the previously-developed σ_w climatologies.”

- Make it clear what the 3 climatologies were. Presumably PI, PD and PD+4K?

(Remarks on code availability)

Version 2:

Reviewer comments:

Reviewer #3

(Remarks to the Author)

I thank the authors for adequately addressing my comments, and recommend the manuscript to be published at this stage.

(Remarks on code availability)

Reviewer #4

(Remarks to the Author)

Please see the attached PDF for the review.

[Editorial Note: See end of file]

(Remarks on code availability)

I have checked that the code is there, but have not looked at it in detail.

Authors' Response to Reviews of

Significant trend in vertical wind velocity variability over the last 40 years

Donifan Barahona, Katherine Breen, Minhgui Diao, Derek Ngo, Flor Vanessa Maciel, Ryan Patnaude
Nature Communications, 2025

RC: Reviewers' Comment, AR: Authors' Response, □ Manuscript Text

We appreciate the thorough analysis of our work and the constructive comments. The primary critique concerns the lack of discussion on the uncertainty in the observational data and the role of MERRA-2 in the estimation process. As we demonstrate below, these critiques do not point to fundamental flaws but rather stem from misunderstandings about our approach. We have thoroughly clarified these points in the revised manuscript, as detailed below.

To enhance readability, we have restructured the paper into two separate documents: the main text and the supplementary material. The supplementary material now includes the content previously found in the Annex, along with newly added plots for clarification and emphasis. The figures in the main text remain unchanged.

The table below maps the numbering of the figures in the original Annex material to their corresponding numbers in the supplementary information of the revised version.

Revised	Original	Remarks
S1	None	New figure
S2	None	New figure
S3	A1	
S4	None	New figure
S5	A2	
S6	A3	
S7	A4	
S8	A5	
S9	A6	
S10	A7	

1. Reviewer #1

1.1. Major concerns

RC: *have three primary criticisms of this article. After these are addressed, I think it will be suitable for publication. I have entered major revisions.*

AR: We thank the reviewer for the positive assessment.

RC: *Comment 1: There needs to be more information on how σ_W actually influences a global model. Presumably this is via the parameterization of the turbulent flux terms? Is σ_W diagnosed in the cumulus parameterization, and in your sensitivity experiments you are simply replacing the diagnosed value with your PI and PD values?*

AR: Yes, we are replacing the diagnosed values with the PI and PD climatologies, but only within the cloud microphysics scheme, where σ_W is used to estimate the supersaturation forcing for aerosol activation. By replacing only these values, we can isolate the effects of σ_W on cloud and ice crystal formation rates, and subsequently study how changes in σ_W influence the evolution of cloud microphysical properties without introducing other confounding factors.

We have clarified the emphasis on cloud microphysics in the revised manuscript, particularly in Section 3 where we dubbed the effect the “cloud microphysics feedback”. We have also changed the title to: “Trends in vertical wind velocity variability reveal cloud microphysical feedback”.

RC: *Comment 2: Fig 2 and the associated discussion would benefit from a correlation analysis. Are the differences in σ_W among the field projects/locations statistically significantly correlated with the analogous differences computed from MERRA-2?*

AR: Figure 2 is not a point-by-point comparison, as it is challenging to precisely determine the aircraft altitude at each time step to collocate it with the MERRA-2 grid. Instead, we compared the overall statistics of σ_W for the entire flight track against those derived from the model, using the same location and altitude range. However, we have conducted a point-by-point comparison in a previous validation of our model [Barahona et al., 2024], using ground-based data that is easier to collocate with the MERRA-2 grid. The reason we prioritized aircraft data in this study is that it is entirely independent of the model training, demonstrating the ability of the B24 model (now termed Wnet) to generalize beyond its training data.

We acknowledge that Figure 2 shows relatively little variability between flights, which may give the impression that the model and observations are uncorrelated. To address this, we have included the new Supplementary Figure S2 that compare σ_W across different environments, including cirrus clouds, the boundary layer, and convective systems, across the globe (see also Table S1 and Figure S1 in the new supplementary document). These comparisons where σ_W spans two orders of magnitude show that the model accurately estimates σ_W across a wide range of atmospheric conditions. We have also added a new paragraph to Section 5.2 detailing the validation against ground-based data.

RC: *Comment 3: The B24 model was derived using a 5 km grid spacing model, which under-resolve a lot of convective motions including those in tropical deep convection, shallow convection, and smaller scale cloud features. What are the implications of this limitation? At least some discussion of this issue is warranted.*

AR: Response: We appreciate this comment. Indeed, a 5 km grid spacing cannot resolve all relevant scales, which we previously demonstrated in our publication detailing the development of the neural network (NN) model [Barahona et al., 2024].

The NN was developed in two main stages: initial training and refinement. The initial training was performed on the 5 km model, upsampled to a half-degree resolution to simulate what a coarse-resolution model like MERRA-2 would capture. This step, known as feature extraction, allows the model to learn the relative spatial variations in σ_W , even though it significantly underestimates its magnitude. In the refinement stage, observational data is used to correct for these biases, using generative training to filter experimental error.

However, ground-based observations alone cannot provide information about spatial variability in σ_W . Therefore, our model derives its ability to predict σ_W across a wide range of conditions from the synergy between high-resolution simulations and observational data.

We have clarified this point in the revised manuscript. The introduction has been extensively modified to better explain our approach and Section 5.1 now includes a new paragraph detailing the development of the neural network.

2. Reviewer #2

RC: *This study relates observed variability in vertical air motion ($\sigma_{W_{obs}}$) to model-parameterized sub-grid-scale vertical air motion ($\sigma_{W_{model}}$) using a deep learning technique. It then uses the global, multi-decade model (the MERRA-2 reanalysis) to predict σ_W globally and over multiple decades. A multi-decade (1980-2023) global trend in σ_W is described. This trend is attributed to changes in the climate system. This trend is intriguing. A Shapley Additive explanations (SHAP) analysis shows that the trend is related to global warming and coincident increases in water vapor mixing ratio.*

AR: Thank you for your constructive comments. We would like to clarify that our technique does not rely on a parameterization of σ_W . Instead, we use explicit simulations of W , which are upsampled to obtain σ_W at half degree horizontal resolution and then refined using observational data.

We have now made this distinction clear in several places in the revised manuscript.

RC: *Assuming that the described trend in σ_W is accurate, one wonders whether this finding is noteworthy. Climate system feedbacks, in particular cloud feedbacks, do involve σ_W , e.g. mid-tropospheric σ_W is largely due to deep convection, so an independent measurement of a σ_W trend would be most noteworthy, as it would explain the observed global increase in precipitation rate (and the rate of hydrological cycling). But this paper does not present an independent observational finding. Rather, since σ_W is parameterized in MERRA-2, the documented trend simply is a dynamically consistent consequence of an increase in global precipitation rate captured by MERRA-2 and other reanalyses. So, I argue that the key finding of this paper lacks novelty. It merely confirms an already established positive trend in convective precipitation.*

AR: We appreciate this comment, as it made us realize that the way the paper was written may have made its scope unclear. σ_W is not parameterized in MERRA-2 or any other reanalysis, hence σ_W from MERRA-2 is not used in the development of the neural network model. We acknowledge that there is a well-established positive trend in convective precipitation, which is related to W , although its relation to a broadening of the spatial distribution of vertical wind velocity is not clear, and it is out of the scope of this work.

Our focus is on how σ_W influences the microphysical properties of clouds. For instance, a broader vertical velocity spectrum leads to more frequent strong droplet formation events, ultimately reducing droplet size. We also examine how these microphysical modifications are systematically driven by large-scale meteorological changes. Our paper specifically addresses the microphysical aspects of clouds rather than the large-scale evolution of convection, offering a novel perspective on the feedback mechanism. To better reflect this focus, we have renamed the article: “Trends in vertical wind velocity variability reveal cloud microphysical feedback”.

RC: *This paper, and references cited, generally provide the details necessary for the method to be followed work and the results to be reproduced. However questions do remain, e.g. what assumptions are made to reduce Doppler hydrometeor vertical velocity (from radar) to vertical air motion. The key reference [20], on which*

this study builds, does not detail these assumptions.

AR: Our study does not generate new experimental retrievals of σ_W . Instead, we refer to the appropriate references that detail the assumptions and approximations involved in its estimation for each case. However, this does not impact our results, as our technique is designed to minimize the effect of errors in the estimated σ_W , as explained below.

RC: *While the paper is well written, the study in-depth, and the findings original, I believe there are flaws in the data analysis, interpretation and conclusions. First, the training dataset of observations have unacknowledged uncertainties or limitations, adding noise to $\sigma_{W\text{obs}}$. Profiling Doppler Lidar (DL) observations are generally limited to the boundary layer (BL) where σ_W is fundamentally different from that in the free troposphere, where the deep learning technique is mostly applied. Doppler radar (DR) data are used as well, but unknown variations in hydrometeor fallspeed contribute significantly to the observed variability, hence the DR variability is a poor proxy for σ_W . Also, DR data are limited to precipitating regions only, while DL are limited to clear air, so their σ_W values represent different atmospheric conditions. Independent airborne estimates of σ_W are used to validate the MERRA-2-based (predicted) σ_W , but the airborne measurements are made elsewhere (above the boundary layer, outside precipitating areas, in clouds that neither DL nor DR can sample, and mostly in the upper troposphere, as indicated by the field campaigns listed in Table 1), and with an unacknowledged uncertainty (especially on the low-frequency side). σ_W values in the upper troposphere typically are much lower than in the BL, so the validation does not capture the spectrum of variability.*

AR: We appreciate the reviewer’s comment, which has helped us clarify the scope of the paper.

Accounting for error is a critical aspect of our approach. We acknowledge that both W and σ_W are derived quantities that carry significant experimental uncertainty. This is now explicitly mentioned in the introduction. If a neural network were trained using standard supervised learning on such data, it would inherit the issues mentioned by the reviewer, including localization, episodicity, and observational uncertainty. However, this is not how our approach works.

To clarify, our model development uses a structured process that we refer to as **Constrained Adversarial Training (CAT)**, shown in Figure A1 (now S3), following the steps:

1. **Training on High-Resolution Simulations:** We first use *global* high-resolution simulations that explicitly compute σ_W to train a neural network (NN) to predict σ_W as a function of the *coarse* meteorological state, denoted as X . The NN is applied globally, not just in the upper troposphere as suggested. Importantly, there is no parameterization of σ_W in the model, i.e., W and σ_W are explicitly simulated.
2. **Freezing Initial Weights:** Once trained, the weights of this NN are **frozen**, meaning these layers remain unchanged during further training with observations.
3. **Adding Correction Layers:** Additional layers are added to the NN to be trained using observational data. These new layers compensate for the fact that the original model cannot resolve all variability in W . The resulting network, which consists of the frozen layers plus the new layers, is termed **Wnet**.
4. **Building a Discriminator:** A second NN, known as the **discriminator**, is introduced to guide training against observational data. This marks the beginning of the **refinement** process.
5. **Training with a Generative Adversarial Network (GAN):** Instead of direct supervised learning, we use a GAN to conduct *unsupervised* training. For this, the discriminator is trained to *learn a structured*

representation of the data, with its input being the observed σ_W . Its goal is to predict σ_W , and in doing so, it filters out observational noise. This step is crucial because neural networks can only learn from structured data by rejecting noise, effectively making the discriminator a **denoising** mechanism.

6. **Refining Wnet:** The GAN algorithm then refines Wnet by training it alongside the discriminator. However, Wnet is **not trained directly on raw data** but rather on the error-filtered representation learned by the discriminator. Additionally, the discriminator penalizes Wnet if it generates noise, forcing Wnet to learn the **true distribution** of σ_W .
7. **Applying to Global Data:** Once Wnet is refined, we use MERRA-2 data to calculate the state X and estimate σ_W globally.

Steps 5 and 6 represent the **breakthrough** of our approach: the use of the GAN's **denoising property** to filter out error, resulting in an NN model that captures the **true physical relationship** between σ_W and the state X . This applies to all **non-systematic errors**, significantly reducing uncertainty in the final model.

Our recent publication Barahona et al. [2024] provides a detailed explanation of these technical aspects. However, we now recognize that by leaving these details to a separate publication, we did not establish a clear connection to this work.

To address this, we have significantly clarified these points in the revised manuscript as follows:

- The introduction has been extensively rewritten to better detail the approach and objectives behind the development of the neural network, particularly in paragraphs 3 and 4.
- A new paragraph has been added to Section 5.1, providing further details on the deep learning approach.
- Throughout the revised manuscript, we have consistently emphasized the denoising properties of our method.

We are confident that these revisions will clarify our approach.

RC: *Second, the frequency spectrum of data from ground-based profiling systems (DR and DL) and that from airborne gust probes differ, and is different from models' sub-grid-scale vertical velocity.*

AR: This is by design. Our goal was to develop a neural network (now termed Wnet) capable of predicting σ_W for all plausible atmospheric conditions. To achieve this, we intentionally incorporated data from different sources—both simulated and measured—obtained using various techniques.

However, a critical aspect of neural network development is ensuring that the model generalizes beyond its training data. For this reason, we placed particular emphasis on comparisons against airborne data, as it is entirely independent of the remote sensing data used to train Wnet. Nevertheless, we also conducted extensive testing of Wnet on a subset of the remote sensing data that was not included in training, as detailed in [Barahona et al., 2024].

In the revised manuscript, we have included a comparison against remote sensing data. This is now detailed in Sections 2 and 5.2, and depicted in Supplementary Figures S1 and S2. The results demonstrate that Wnet accurately predicts σ_W across a wide range of conditions, including cloudy layers, clean air, and along flight paths.

RC: *Third, I question the robustness of the key finding, that in the last four decades, the global troposphere has experienced an increase in σ_W of about 1% per decade, reaching up to 1% per decade in some*

regions. This is according to the MERRA-2 dataset, subjected to a ML technique trained on data that have measurement uncertainty and lack representativeness. The impact of measurements uncertainties and excessive clustering in the physical and the spectral domains is not analyzed.

AR: We are confident that the responses above demonstrate that our estimate is unbiased, does not suffer from a lack of representativeness since we train on a wide range of conditions, and is not subject to excessive clustering, as the initial training is conducted on global data while the remote sensing and airborne datasets cover most regions of the world (see the Supplementary Figure S1).

This follows from our two-stage training approach. The initial training, performed using a global storm-resolving model, allows the neural network to learn the relative global spatial variation in σ_W , even if it underestimates its absolute value. In the refinement stage, observational data is incorporated to correct for these biases, with generative training accounting for error. However, ground-based observations alone cannot provide information about the spatial variability of σ_W . Instead, the synergy between high-resolution simulations and observed data, while accounting for experimental error, enables our model to predict σ_W across a wide range of atmospheric conditions.

RC: *Fourth, MERRA-2 is one reanalysis. The same technique was used on one other reanalysis, ERA5, and the results are quite different (Fig. A3): the trend essentially vanishes above the boundary layer (500 and 250 hPa) in ERA5, and the spatial patterns across the globe are different (Fig. 1 vs Fig. A3). The differences in values between ERA5 (monthly-mean data) and MERRA-2 (6 hourly) makes sense: monthly-mean data do not capture the synoptic variability in σ_W . But that does not explain the differences in trend. Given this disagreement, I would recommend the use of full-time-resolution ERA5 data, and other global or regional reanalyses, to build further evidence.*

AR: Thank you for your comment. This was an oversight on our part—we should not have used monthly means.

In the revised version of the manuscript, we have instead used instantaneous data from ERA5. The updated results, shown in Supplementary Figure S6, are much more consistent with MERRA-2 and further reinforce our conclusion of an intensification of subgrid-scale variability over the past four decades. These results are thoroughly discussed in Section 2 of the revised manuscript.

RC: *Some of the terminology used in the paper is unfortunate and inconsistent with expected standards. For instance, vertical “wind” is used, where the term wind refers to horizontal air motion. I acknowledge that other papers have used the expression “global cloud resolving models”, but of course, global models cannot truly resolve cloud processes: the latest DNS models, which capture cloud microphysical processes, have meter-scale volumes at most. I think they refer to global convection-permitting models, which exist, e.g. the E3SM SCREAM. But then, MERRA-2 is not one of them. Maybe they simply refer to numerical models in which cloud processes are parameterized. That has been done in climate models since the beginning, going back to the early CMIPs. Statements like “cloud formation occurs at scales around 100 m” depart from the established core knowledge, in this case understanding of the multiscale nature of clouds. There is no “Pacific Cold Tongue at 250 hPa”. There are many other examples.*

AR: Vertical velocities from MERRA-2 were not used for training. The B24 model (now termed Wnet) serves as a link between the coarse atmospheric state and microscale variability. It was trained using results from a non-hydrostatic model with a 7 km horizontal resolution that explicitly computes W (with upsampling to a half-degree resolution to obtain σ_W). We acknowledge that a 7 km resolution does not fully capture the entire spectrum of W , which is why we introduced the refinement step. To ensure clarity, we have thoroughly revised the manuscript to improve the terminology.

References

Donifan Barahona, Katherine H Breen, Heike Kalesse-Los, and Johannes Röttenbacher. Deep learning parameterization of vertical wind velocity variability via constrained adversarial training. *Artificial Intelligence for the Earth Systems*, 3(1):e230025, 2024. .

Authors' Response to Reviews of

Trends in vertical wind velocity variability reveal cloud microphysical feedback

Donifan Barahona, Katherine Breen, Minghui Diao, Derek Ngo, Flor Vanessa Maciel, Ryan Patnaude
Nature Communications, 2025

RC: Reviewers' Comment, AR: Authors' Response, □ Manuscript Text

We sincerely appreciate the time and effort invested in reviewing our work. Below, we provide a detailed response to each of the concerns raised.

1. Reviewer #3

RC: *In the manuscript “Trends in vertical wind velocity variability reveal cloud microphysical feedback” Barahona et al. apply their recently published generative AI model (Wnet) to predict the spatial variability (standard deviation) of vertical wind W from large scale meteorological conditions from reanalysis (MERRA-2 and ERA-5). They find significant increases in W that are particularly strong over low (900hPa) and mid-level (500hPa) oceanic regions. This finding suggests increasing variability in supersaturation and therefore changes in aerosol activation and cloud microphysics, and implies a (comparably small) negative feedback. This finding is important and justifies the publication in Nature Communications. My main task as a reviewer was to evaluate the authors response to the criticism by reviewer 2 (R2), who, in her/his review was very skeptical about the methodological approach, i.e. about the ability of the Wnet to accurately predict W. After reading the manuscript, the reviews and the authors responses, I am convinced that the criticism of R2 is mainly founded on an incomplete understanding of the details of the generative AI method developed by the authors. In my opinion, the authors have done substantial work to improve the communication and clarity of the manuscript in this regard.*

AR: We thank the reviewer for the positive assessment on our work.

RC: *I do question the authors focus on one reanalysis to predict W, though. In the manuscript, the results of W data generated with the large scale fields from MERRA-2 are highlighted, I guess because the trend signal is much more pronounced than in ERA-5. In my opinion this is not adequate and seems a bit like cherry picking of the results to me unless there is a physical reason why we should trust MERRA-2 more than ERA5. If that is not the case I believe the two results should be equally weighted in the interpretation and in the headline statements. In this aspect, I agree with R2, comment 4, and think it is not appropriate to overrepresent the findings from the reanalysis that produces the stronger trend signal in σ_W . I acknowledge that ERA5 is now discussed more in the manuscript than initially, but still there seems to be a focus on the aspects where the two data sets agree rather than clearly state the differences between them.*

AR: We agree that using two reanalyses instead of relying solely on MERRA-2 makes the results more robust. Our initial choice was not about selectively presenting results, but rather due to the practicality of performing a long-term, high temporal resolution calculation of σ_W using the large ERA5 dataset compared to MERRA-2, which is readily stored on NASA servers.

To address the reviewer's concern, we have now extended the calculation of σ_W using ERA5, increasing the temporal resolution to 3 hours and using all 37 pressure levels over the full research period. This enabled a closer comparison between MERRA-2 and ERA5 results and allowed us to evaluate both datasets against observations. We now report our main results as the average of both estimates. Sections 2 and 3 of the revised manuscript discuss differences between the reanalyses, and the individual results are presented in the supplementary figures.

2. Reviewer #4

2.1. Responses to the comments of the other reviewers.

Reviewer 1

RC: *There was some confusion (comment1) about what the ERF that was calculated represented, but I think that the main manuscript has been changed to make it clear now that the ERF value is just due to the microphysical effect on droplet/ice activation. However, this is still not the case in the abstract and I have made some suggestions for that below.*

AR: Thanks. We answer these below.

RC: *I think that comment2 has been addressed reasonably.*

AR: Thanks for taking the time to verify it.

RC: *Comment3 concerns the fact that the 5km resolution of the high resolution model won't resolve a lot of the relevant motions and suggests more discussion. I agree with this point. Some discussion has been added, but I think that a little more discussion would be helpful. I made some similar comments below for Section 2., which are repeated here:- - Here it would be good to discuss that there is a large range of scales of updrafts/turbulence (from global circulations, though to deep convection, shallow convection, stratocumulus, small-scale mixing and turbulence). And how sub-grid σ_W will depend on the resolution being considered – i.e., what is the size of the “grid”. In this study you use a 5km model, but also include some smaller scales with the observations – what resolutions/scales do the observations used to build the model cover? And what is the resolution of the aircraft observations used to test the model?*

AR: All observations of W , ground and aircraft-based, are retrieved with a frequency between 1 and 10 Hz, corresponding to a spatial scale between about 30 m and 300 m [Kalesse and Kollias, 2013, Newsom et al., 2019], covering most of the ranges of atmospheric motion. From these σ_W is estimated using a characteristic time of 7 minutes for aircraft measurements and 5-30 minutes for ground based so that σ_W represents variability over a 0.5 to 1 degree grid cell. There is little variability in W resolved in the 25-50 km range (and above 50 km) so we still apply W_{net} to the ERA5 reanalysis, with the caveat that it might lead to slight underestimation. We do not recommend W_{net} for finer grids without proper scaling, although our 7 km nature runs shows that a lot of variability is still unresolved at that scale (hence our usage of observations directly).

We have expanded the methods section clarifying this point.

RC: *The results for the ERF calculation rest on changes in stratocumulus clouds in the sub-tropics, off the west coasts of S America, N America and SW Africa. However, there is little discussion about the spatial scales of vertical winds in these regimes and about how well your model captures them. One approach used in the past, which should be discussed in this paper, has been to use very high resolution (35-150m) LES models to quantify σ_W at stratocumulus-relevant scales (<https://agupubs.onlinelibrary>).*

wiley.com/doi/full/10.1002/2013JD021218). Do the observations you used to build and test the model cover these smaller scales for stratocumulus clouds?

AR: Yes, W is retrieved at spatial resolutions (assuming ergodicity) of the order of tenths of meters. This is enough to resolve most atmospheric motion. To emphasize this point we have added a separate section on validation where we explain our approach more thoroughly. We have also added more information to Tables S1 and S2 (of the revised work) detailing the environment and cloud regimes associated with each observational dataset.

Thanks for pointing the LES work to us. We now add this statement to the introduction as well: Similarly σ_W can be explicitly resolved using Large Eddy Simulations (LES) which have been used to develop parameterizations for climate modeling, mostly focusing on stratocumulus regimes [Malavelle et al., 2014]. LES are however limited in domain and it is not clear to what extent these results are applicable across the wide diversity of meteorological environments present in the atmosphere.

Reviewer 2

RC: *Some of the comments may have come from unclear explanations in the introduction and methods sections and there has been some effort to improve that. However, this aspect could still be improved further and I've tried to make some suggestions for that below. It is still not clearly stated whether the sub-grid parameterized σ_w of the high resolution model is used along with the resolved updrafts (I don't think it is, but I'm not sure that it's stated clearly).*

AR: There is no parameterization of σ_W in the GEOS 5 km model, nor in MERRA-2; σ_w comes from the coarsening of W from the high resolution model to half degree. For the initial training of Wnet, σ_w is calculated using 64 values of W per each half-degree grid cell. This leads to a σ_w too low against observations, which we correct in the second stage of training. This step however tells Wnet how to extrapolate between different conditions and locations, as detailed in our previous work.

This is now clearly stated in the methods section.

RC: *The comment "While the paper is well written, the study in-depth, and the findings original, I believe there are flaws in the data analysis, interpretation and conclusions.", also questions the types of regime sampled by the remote sensing used to build Wnet and the aircraft obs used to test it. I agree with this aspect – it's not clear which types of cloud/meteorological regimes have been sampled by the aircraft observations from the tables and text provided. There is some useful information in Table S1 for the ground station data, but more information on the cloud types sampled would be useful – does "PBL" cover stratocumulus and trade cumulus, or just clear-air observations? Nor is it clear what the resolution of the observations was. Some more discussion on this and the limitations is warranted (linking to the comments above about stratocumulus). I think the other comments have been addressed well.*

AR: Thanks for the assessment. We carry out a rigorous testing of the results, covering a wide range of meteorological environments and cloud regimes. We however agree that it was not properly conveyed in the manuscript.

To address the reviewers concern we have expanded the explanation of the relevant scales in Section 2 and 3. In particular we have added the following paragraph to the new Section 2.1 on validation:

"In situ measurements and ground-based observations are used to validate the climatologies of σ_W . Since σ_W is generally not measured directly, it is derived from high-frequency (typically 1–10 Hz) measurements of W [Maciel et al., 2023, Newsom et al., 2019], corresponding to spatial scales of roughly 30–300 m, which encompass most vertical motions relevant for cloud formation. To calculate σ_W , the W fields are coarsened

using a characteristic time scale chosen so that σ_W represents the standard deviation of W over a spatial scale of approximately 50–100 km, making it comparable to that of a GCM grid cell. For aircraft measurements, σ_W was calculated using a 430 s averaging window, representing a horizontal scale of $\Delta x \sim 100$ km [Maciel et al., 2023], covering both in-cloud and clear-sky regions. For ground-based retrievals, the characteristic time scale depends on the mean horizontal wind at each site and ranges from 15 to 30 minutes, corresponding to $\Delta x \sim 50$ km [Barahona et al., 2024]. Observations were collected from multiple locations worldwide, spanning a wide range of cloud regimes (cirrus, stratocumulus, altocumulus, convective) and employing various measurement techniques for W , providing an unbiased and independent validation of σ_W .”

We have also expanded the tables summarizing the campaigns and ground based sites (now tables S1 and S2), detailing the environment and cloud regimes.

2.2. My own comments

Major Comments

RC: *One of the key results (the “ERF” forcing due to σ_W changes) rests on the difference in the σ_W climatologies in the PD run compared to that from the coupled PI run being representative and not just a random chance result. E.g., if different 10 year samples were chosen or if longer samples were used, would the results be the same? I don’t think that 10 years is long enough to capture the variability that is possible in a coupled simulation such as the PI simulation. In a coupled run the SST patterns, sea-ice amount, etc. will vary a lot and 10 years won’t capture this. Can you use the full 180 years? Even for the prescribed SST runs it is too short since there will be a lot of natural variability in the climate that you would need to sample over. The exact 10-year period chosen may also influence the mean climatological winds, circulation, etc. and it’s possible that different 10-year periods might give different results. I would expect a longer period to be used (around 30 years). It would be good to compare results using 10, 20, 30 years to make sure they are robust. And similarly in the coupled PI run, although you will probably have to extend to even longer periods for that. Possibly related to the comment above is the concern that the differences in σ_W between the PD and PI runs don’t seem consistent with those between the +4K and PD runs (Fig. S7). There is also little resemblance to the long-term trends. This is a concern as it suggests that the impact of warming and a changing climate is not being captured consistently with the model experiments. There are some more detailed comments on this below, but the longer experiments above should help to address the issue. More discussion on these discrepancies in the results is needed.*

AR: We appreciate these comments, as they prompted us to reconsider our approach. In our original calculations, we attempted to isolate the effect of σ_W on cloud microphysics using previously run simulations. While this approach was considered reasonable at the time, we agree with the reviewer that it is problematic for the reasons outlined. Addressing these issues through a GCM-based approach would require greatly expanding the simulations, which would be challenging and still subject to uncertainties. Instead, we have overhauled our radiative forcing calculation, adopting methods commonly used in satellite-based estimations that have proven successful in the literature. In summary, we have:

- Calculated pre-industrial σ_W using the ERA-20C reanalysis, which is constrained by records of SST, winds, and surface water vapor. We used the period 1901–1905, the earliest available in the dataset.
- Focused solely on the effect of σ_W on low-level clouds, where its influence is expected to be dominant, we extended a well-known relationship used to study microphysical effects on shortwave forcing [Yuan et al., 2024]:

$$\Delta SW = -SW_{\text{down}} C_f A_c (1 - A_c) \frac{1}{3} \frac{\partial \ln N_d}{\partial \ln \sigma_W} \Delta \ln \sigma_W \quad (1)$$

where SW_{down} is the downwelling shortwave flux at the top of the cloud layer (assumed here to be equal to SW at the surface [Toll et al., 2019]), C_f is the liquid cloud fraction, and A_c is the cloud albedo.

Although less sophisticated than a full GCM simulation, we consider this approximation more intuitive, robust, and consistent with the data-driven, model-independent nature of our work.

We also conducted a detailed sensitivity analysis of ΔSW with respect to each factor in the above expression. While we acknowledge the limitations of this approach, such as neglecting liquid water path and cloud fraction adjustments, as well as the effects of σ_W on ice clouds, we expect these to be minor given the modest influence of σ_W on N_d .

Section 3 has been fully revised to describe our updated approach.

Detailed comments

***** Abstract

RC:

- *“This effect is however difficult to represent” - Better without the “however” here since it is also used later in the sentence.*
- *“however can be addressed by machine learning.” -> however, it can be addressed...*
- *“indicating enhanced turbulent atmosphere” -> indicated enhanced atmospheric turbulence?*
- *“suggesting a feedback connection between enhanced warming, turbulence and cloud microphysics, with an associated radiative forcing of about $0.1 \pm 0.21 \text{ W m}^2$, slightly mitigating greenhouse warming.” - You haven’t introduced the effect on microphysics here yet and it’s also not clear here that the “forcing” is solely due to the microphysical effect. I suggest something like this – “suggesting a feedback connection between enhanced warming and turbulence, which in turn has a microphysical effect through the activation of more cloud hydrometeors. The forcing associated with the microphysical effect is estimated here to be about $0.1 \pm 0.21 \text{ W m}^2$, slightly mitigating greenhouse warming.”*

AR: Thank you for the suggestions. They have been incorporated in the text.

***** Introduction

RC: *“modern atmospheric models” – You should specify the subset of models being talked about here rather than later in the paragraph - this only applies generally to global models. E.g., LES models have much higher resolution.*

AR: The statement now reads: “modern global atmospheric models”

RC: *“Nevertheless, the GSRM simulations and the extensive data they produce represent a wealth of information that can be used to understand and model W. Artificial intelligence techniques and data-driven algorithms like deep learning, are well-suited for this task. Unlike traditional numerical simulations or theories, these algorithms learn from data to make predictions.” – You haven’t introduced observational data yet – surely this is necessary to allow the AI approach to improve the predictions of the GSRM simulations?*

AR: The statement now reads “The GSRM simulations and the extensive data they produce represent a wealth of information that can be used to understand and model σ_W , particularly when combined with observational

datasets.”

RC: *“it was trained on results from global storm-resolving model simulations,” - It would be helpful to quote the resolution of the model here to make it clear that it is built on high-resolution data. I don’t think that it is mentioned in the whole paper. Also, to avoid confusion later, it would be good to clarify in this sentence that it is trained to predict σ_W as output from the high-resolution model as a function of the low resolution meteorology (was this from MERRA or from the coarsened fields of the high-res model?)*

RC: *“spatial distributino” typo. “By integrate physical relationships” -> By integrating...*

AR: Corrected.

RC: *“Since the Wnet was built using data from reanalyses like MERRA-2,” - Was it built from MERRA2 or from coarsened fields from the high-res model? If the latter then it would perhaps be better to say that the Wnet model uses input variables that can provided by MERRA2.*

AR: The statement has been modified to: “ Since the Wnet was built to use variables that can be provided by reanalyses, it is possible to reconstruct σ_W globally using their output.”

RC: *“Our analysis revealed” - Better to say “We will present an analysis that shows” or similar since this sounds like you are referring to your past study.*

AR: The statement has been modified as: “We will present an analysis showing significant trends in σ_W emerging over recent decades, with implications not only for cloud formation but also climate change assessments, and weather and turbulence forecast.”

***** *Results*

RC: *“We used the Wnet model to create three-dimensional global distributions of W every 3 hours from 1980 to 2023.” - it would be good to say that you used the meteorological fields from the 1980-2023 MERRA2 reanalysis as inputs into Wnet to create the climatology. Better here than in the next sentence. Fig. 1 caption - “Stippling highlights statistical significance the 95% level” -> “... at the 95% level.*

AR: Corrected.

***** *Section 2.1*

RC: *- Here it would be good to discuss that there is a large range of scales of updrafts/turbulence (from global circulations, though to deep convection, shallow convection, stratocumulus, small-scale mixing and turbulence). And how sub-grid σ_W will depend on the resolution being considered – i.e., what is the size of the “grid”. In this study you use a 5km model, but also include some smaller scales with the observations – what resolutions/scales do the observations used to build the model cover? And what is the resolution of the aircraft observations used to test the model?*

AR: This comment is addressed in the response to the major comments.

RC: *The results for the ERF calculation rest on changes in stratocumulus clouds in the sub-tropics, off the west coasts of S America, N America and SW Africa. However, there is little discussion about the spatial scales of vertical winds in these regimes and about how well your model captures them. One approach used in the past, which should be discussed in this paper, has been to use very high resolution (35-150m) LES models to quantify σ_W at stratocumulus-relevant scales (<https://agupubs.onlinelibrary.wiley.com/doi/full/10.1002/2013JD021218>). Do the observations you used to build and test the model cover these smaller scales for stratocumulus clouds?*

AR: They do. Sites like like ENA [Rémillard et al., 2012], NSA [Verlinde et al., 2016] and ASI [Zuidema et al., 2018] are typically covered with low level stratocumulus. Wnet performs very well at those conditions (see new Figure 2).

We have expanded the Section to clarify our approach to the testing and validation, detailing the scales and regimes involved. We also refer to the LES approach in the introduction.

***** Section 2.2

RC: *“depicted with in Figure 1.” – typo. “frequency small scale processes” -typo*

AR: Corrected.

***** Section 3

RC: *“However, while turbulence indices are derived from the coarse meteorological state, σ_W indicates variability at the small scale,” - it’s not very clear what you mean here.*

AR: The statement has been removed

RC: *“corresponds to a previously unexplored feedback mechanism mediated by the microphysical properties of clouds and the small scale dynamics.” - Although this will be inherently included in GCMs in some way since many predict TKE as a function of the grid properties, which then affects cloud droplet activation.*

AR: In principle, yes, but it is unclear how accurately current models reproduce observed σ_W or its global trends. Moreover, the role of σ_W as a feedback mechanism linking large-scale dynamics and cloud microphysics has not been isolated nor thoroughly explored.

We have reworded the statement as:

“While it is well established that large-scale cloud properties such as cloud fraction and liquid water path (LWP) are sensitive to changes in SST and water vapor shifts, their impact on microphysical properties like cloud droplet number and size, remains largely unexplored. This represents a potential feedback mechanism mediated by the microphysical properties of clouds and small-scale dynamics.”

RC:

- *“We employ global atmospheric modelling” - should say which model in the main text.*
- *“Near the surface, the difference between the PD and PI climatologies resembles the anomalies seen in Figure 1.” - I can’t see any similarity between Fig. 1f and the lower-middle plot in Fig. S7 (900 hPa PD-PI) if that’s what you’re referring to. Nor for 500hPa. Plus, the PD-PI plots have the opposite sign to the SST+4K - PD plots. I would trust the latter a bit more since both are driven by SSTs and don’t rely on the 10-year coupled climatologies. The signals are certainly much stronger at higher altitudes.*
- *“Near the surface, the difference between the PD and PI climatologies resembles the anomalies seen in Figure 1.”- Why would a decrease in cloudiness lead to less convective transport of water. Where does the cloudiness decrease?*
- *“Enhanced W in these regions leads” - Which regions? Low-cloud stratocumulus regions? Which ones in particular? I can’t really see any enhanced σ_W in these regions in Fig. S7? “Figure S9). In contrast, extra-tropical regions in the Pacific experience the opposite effect. Drying at lower levels*

weakens W as convective clouds deepen in the Tropics [37]. A sensitivity run using a W climatology generated from a scenario with present-day SSTs increased by 4 K [43] suggests that as drying intensifies, W weakens even further,”

- *Figs. 4 and S10 - It seems that the SWCE and LWCE changes should be negated to make them net incoming (instead of outgoing) TOA values for consistency with the ERF (since the value in Fig. 4 is negative giving an overall cooling)? And the definition of ERF in the caption is wrong – it should be net incoming TOA radiative forcing?*
- *“the radiative forcing obtained after 50 years of integration.” - so do you use the 10 year σ_W climatology repeated 5 times? You should make it clearer what you did.*
- *- The change in low cloud cover must come from the change in N_d since σ_W only affects the micro-physics. And so the reduction in cloud cover probably comes from the precipitation enhancement as a result of the reduced N_d ? It would be good to mention this.*

AR: We greatly appreciate the reviewer’s analysis. In our original approach, we used 1-year σ_W climatologies, obtained by averaging the input state from a previous simulation over ten years and then running a new simulation repeating the climatology 50 times. While this was considered sound, we acknowledge that it relied on a single model and a single set of simulations, making it inherently uncertain.

We have since realized that, in attempting to be as rigorous as possible, we may have introduced additional uncertainty. The individual effects of many variables are difficult to isolate using GCM simulations unless the modeling framework is greatly expanded, for example by employing large ensembles or long-term coupled runs.

Even with such an expansion, extracting a clear signal would remain challenging. Therefore, we have reworked the section to focus exclusively on the effect of σ_W on low-level clouds, as explained in the major comments section. We have also incorporated the reviewer’s feedback where applicable.

RC: *“could profoundly impact climate predictions.” - This seems a bit of a stretch for a 0.2 W/m2 forcing with lots of uncertainty! The language needs modifying here.*

AR: The statement has been removed.

*****Section 5.1

RC: *“trained using long-term retrievals of” - would be good to reiterate that these are observations.*

AR: The statement has been corrected.

*****Section 5.2

RC: *“ranging from 20 seconds to 5 minutes,” - Is it possible to give a horizontal resolution? E.g., for a 10 m/s horizontal wind speed this would be a resolution of 200-3000m.*

AR: In the original description paper [Barahona et al., 2024] this refers to the averaging time to obtain σ_W . The frequency of W retrievals is much higher (1–10 Hz), corresponding to a horizontal resolution of approximately 30–300 m.

We acknowledge it is a misleading statement. It has been modified to:

“To correct for this, a refinement step was introduced, where W_{net} was further trained using long-term observations, which derive σ_W from high frequency radar and lidar retrievals of W ($\sim 1 - 10$ Hz with a

corresponding spatial resolution $\Delta x \sim 30 - 300\text{m}$), covering most atmospheric motion relevant to cloud formation.”

***** Section 5.3

RC:

- **“the MERRA-2 reanalysis and the free-running simulation.” - Make it clearer what these two are - is the reanalysis the same as the "long-term MERRA climatology run" and is the "free-running" simulation the prescribed SST PD run?**
- **“, taken from each of the previously-developed W climatologies.” - Make it clear what the 3 climatologies were. Presumably PI, PD and PD+4K?**

AR: These statements have been removed as the section has been modified.

References

- Heike Kalesse and Pavlos Kollias. Climatology of high cloud dynamics using profiling ARM Doppler radar observations. *Journal of climate*, 26(17):6340–6359, 2013.
- RK Newsom, Chitra Sivaraman, TR Shippert, and LD Riihimaki. Doppler Lidar vertical velocity statistics value-added product. Technical report, DOE ARM Climate Research Facility, Washington, DC (United States), 2019.
- Florent F. Malavelle, Jim M. Haywood, Paul R. Field, Adrian A. Hill, Steven J. Abel, Adrian P. Lock, Ben J. Shipway, and Kirsty McBeath. A method to represent subgrid-scale updraft velocity in kilometer-scale models: Implication for aerosol activation. *Journal of Geophysical Research: Atmospheres*, 119(7):4149–4173, 2014. . URL <https://agupubs.onlinelibrary.wiley.com/doi/abs/10.1002/2013JD021218>.
- Flor Vanessa Maciel, Minghui Diao, and Ryan Patnaude. Examination of aerosol indirect effects during cirrus cloud evolution. *Atmospheric Chemistry and Physics*, 23(2):1103–1129, 2023. .
- Donifan Barahona, Katherine H Breen, Heike Kalesse-Los, and Johannes Röttenbacher. Deep learning parameterization of vertical wind velocity variability via constrained adversarial training. *Artificial Intelligence for the Earth Systems*, 3(1):e230025, 2024. .
- Tianle Yuan, Hua Song, Lazaros Oreopoulos, Robert Wood, Huisheng Bian, Katherine Breen, Mian Chin, Hongbin Yu, Donifan Barahona, Kerry Meyer, et al. Abrupt reduction in shipping emission as an inadvertent geoengineering termination shock produces substantial radiative warming. *Communications Earth & Environment*, 5(1):281, 2024. .
- Velle Toll, Matthew Christensen, Johannes Quaas, and Nicolas Bellouin. Weak average liquid-cloud-water response to anthropogenic aerosols. *Nature*, 572(7767):51–55, 2019. .
- Jasmine Rémillard, Pavlos Kollias, Edward Luke, and Robert Wood. Marine boundary layer cloud observations in the azores. *Journal of Climate*, 25(21):7381 – 7398, 2012. . URL <https://journals.ametsoc.org/view/journals/clim/25/21/jcli-d-11-00610.1.xml>.

J. Verlinde, B. D. Zak, M. D. Shupe, M. D. Ivey, and K. Stamnes. The arm north slope of alaska (nsa) sites. *Meteorological Monographs*, 57:8.1 – 8.13, 2016. . URL <https://journals.ametsoc.org/view/journals/amsm/57/1/amsmonographs-d-15-0023.1.xml>.

Paquita Zuidema, Arthur J. Sedlacek III, Connor Flynn, Stephen Springston, Rodrigo Delgadillo, Jianhao Zhang, Allison C. Aiken, Annette Koontz, and Paytsar Muradyan. The ascension island boundary layer in the remote southeast atlantic is often smoky. *Geophysical Research Letters*, 45(9):4456–4465, 2018. . URL <https://agupubs.onlinelibrary.wiley.com/doi/abs/10.1002/2017GL076926>.

Authors' Response to Reviews of

Trends in vertical wind velocity variability reveal cloud micro-physical feedback

Donifan Barahona, Katherine Breen, Derek Ngo, Flor Vanessa Maciel, Ryan Patnaude, and Minhgui Diao
Nature Communications, 2025

RC: *Reviewers' Comment*, AR: Authors' Response, □ Manuscript Text

We appreciate the time and effort invested in reviewing our work. Below, we provide a detailed response to each of the concerns raised.

1. Reviewer #4

RC:

1. *“Analyzing long-term trends in such datasets help us understand” -> “Analyzing long-term trends in such datasets helps us to understand”*
2. *Fig. 1 – (b) and (d) labels for 2nd column are wrong.*
3. *p.10 – “maybe” -> “may be”*
4. *p.11, paragraph 3 – the sigma needs a w subscript.*
5. *p.11 – “This is explained as a shallow PBL may reflect enhanced stability”. - Better as “An explanation for this is that a shallow PBL is associated with enhanced stability”*
6. *p. 13 – “This strongly suggest” -> “This strongly suggests”*
7. *p.13 – “There is still discrepancy on the estimates of water vapor” -> “There are still discrepancies in the estimates of water vapor”*
8. *p.13 – “there is still some residual error in the approximations made in its development, as for example collocating observational data on a GCM grid.” -> “there is still some residual error in the approximations made in its development, for example in the collocation of observational data onto a GCM grid.”*

AR: All corrections have been incorporated.

RC: *“To calculate σ_W , the W fields are coarsened using a characteristic time scale chosen so that σ_W represents the standard deviation of W over a spatial scale of approximately 50–100 km,” - Coarsened makes it sounds like you are degrading the resolution of the W observations to 50-100km? But I think you mean that you use the native high resolution of the observations and then calculate σ_W over 50-100km boxes (so that the result is a coarse resolution σ_W field)?*

AR: The statement has been rewritten as “To calculate σ_W , the high-resolution W fields are used to compute standard deviations over spatial windows corresponding to a characteristic scale of approximately 50–100 km, representative of variability on scales comparable to a typical GCM grid cell.”

RC: *Fig. 4 – the text on p. 11 and the caption for Fig. 4 mention the beta parameter. However, this is used before it is defined.*

AR: The statement has been modified to: “Uncertainty in ΔSW is estimated from a sensitivity analysis varying the vertical level for σ_W , present day and beginning of the century years, and the reanalysis dataset (see Table S3).” Also the the definition of beta has been added to the caption of Figure 4.

RC: *p.11 – it’s a little unclear about how $\frac{\partial \ln N_d}{\partial \ln \sigma_W}$ is calculated from MERRA2. Does MERRA2 predict N_d and then you do some a regression? Or is there a parameterization for N_d that you use directly? Ok, I see it is in the Methods section – you could refer the reader there for more details perhaps when you talk about it on p.11.*

AR: The paragraph has been rewritten as: “Besides the relative change in σ_W , ΔSW is also controlled by the susceptibility of N_d to σ_W , ($\beta = \frac{\partial \ln N_d}{\partial \ln \sigma_W}$), estimated over the period 1980–2022 by allowing σ_W to vary each year, while N_d is calculated using annually repeating mean climatological aerosol fields from MERRA-2, as described in the Methods section [Yuan et al., 2024].”

RC:

- 1. Top of p.12 – it would be good to add the caveat somewhere that changes in the PBL height (and other meteorological aspects) that are associated with the σ_W changes will be likely to have impacts on cloud cover and thickness that are likely to have large SW effects – these may counteract or enhance the effects of changing N_d .*
- 2. p.12 “For instance, circulation changes that reduce PBLH may lower σ_W , thin clouds, increasing local temperatures, which tends to deepen of the PBL, hence introducing a negative feedback on climate.”. - Better as “For instance, circulation changes that reduce PBLH may lower σ_W and cause a thinning of clouds, which would increase local temperatures. This would then tend to deepen of the PBL, hence introducing a negative feedback on climate.” - Although as mentioned above, there would also be changes in the bulk cloud amount (fraction and thickness) which would affect any feedback from σ_W .*

AR: We agree that changes in PBLH are likely to have a stronger influence on cloud fraction and other large-scale cloud properties, which in turn affect radiative forcing. However, these changes are not primarily driven by cloud microphysics, which is the focus of this study. We have clarified this point in the revised paragraph, where we also incorporated the suggested edits:

“For instance, circulation changes that reduce PBLH may lower σ_W and cause a thinning of clouds, which would increase local temperatures. This would then tend to deepen the PBL, hence introducing a negative feedback on climate. Conversely, if the PBL deepens due to increased SSTs, both σ_W and cloud optical depth may increase, which would tend to cool the surface. Importantly, changes in PBLH can also affect cloud fraction and thickness through non-microphysical processes, which may either offset or amplify the microphysical component of the cloud feedback.”

RC: *p.13 – “coarsened to calculate W over a 50 km grid cell” and “The W fields were coarsened using the average wind at each site to represent” – Is coarsened is the correct word here? Or did you calculate σ_W over 50km grid cells using the 6km w (or obs) data?*

AR: The statement has been rewritten as:

First, the neural network, termed Wnet, was trained using W output from a global storm-resolving simulation

($\Delta x \sim 6$ km), with σ_W computed over areas corresponding to ~ 50 km grid cell.

References

Tianle Yuan, Hua Song, Lazaros Oreopoulos, Robert Wood, Huisheng Bian, Katherine Breen, Mian Chin, Hongbin Yu, Donifan Barahona, Kerry Meyer, et al. Abrupt reduction in shipping emission as an inadvertent geoengineering termination shock produces substantial radiative warming. *Communications Earth & Environment*, 5(1):281, 2024. .

Review of Barahona

“Analyzing long-term trends in such datasets help us understand” -> “Analyzing long-term trends in such datasets helps us to understand”

Fig. 1 – (b) and (d) labels for 2nd column are wrong.

“To calculate σW , the W fields are coarsened using a characteristic time scale chosen so that σW represents the standard deviation of W over a spatial scale of approximately 50-100 km,”

- Coarsened makes it sounds like you are degrading the resolution of the W observations to 50-100km? But I think you mean that you use the native high resolution of the observations and then calculate σ_w over 50-100km boxes (so that the result is a coarse resolution σ_w field)?

p.10 – “maybe” -> “may be”

Fig. 4 – the text on p. 11 and the caption for Fig. 4 mention the beta parameter. However, this is used before it is defined.

p.11, paragraph 3 – the sigma needs a w subscript.

p.11 – it’s a little unclear about how $d\ln(N_d)/d\ln(\sigma_w)$ is calculated from MERRA2. Does MERRA2 predict N_d and then you do some a regression? Or is there a parameterization for N_d that you use directly? Ok, I see it is in the Methods section – you could refer the reader there for more details perhaps when you talk about it on p.11.

p.11 – “This is explained as a shallow PBL may reflect enhanced stability”.

- Better as “An explanation for this is that a shallow PBL is associated with enhanced stability”

Top of p.12 – it would be good to add the caveat somewhere that changes in the PBL height (and other meteorological aspects) that are associated with the σ_w changes will be likely to have impacts on cloud cover and thickness that are likely to have large SW effects – these may counteract or enhance the effects of changing N_d .

p.12 “For instance, circulation changes that reduce PBLH may lower σW , thin clouds, increasing local temperatures, which tends to deepen of the PBL, hence introducing a negative feedback on climate.”.

- Better as “For instance, circulation changes that reduce PBLH may lower σW and cause a thinning of clouds, which would increase local temperatures. This would then tend to deepen of the PBL, hence introducing a negative feedback on climate.”
- Although as mentioned above, there would also be changes in the bulk cloud amount (fraction and thickness) which would affect any feedback from σ_w .

p. 13 – “This strongly suggest” -> “This strongly suggests”

p.13 – “There is still discrepancy on the estimates of water vapor” -> “There are still discrepancies in the estimates of water vapor”

p.13 – “there is still some residual error in the approximations made in its development, as for example collocating observational data on a GCM grid.” -> “there is still some residual error in the approximations made in its development, for example in the collocation of observational data onto a GCM grid.”

p.13 – “coarsened to calculate σ_w over a ~ 50 km grid cell” and “The w fields were coarsened using the average wind at each site to represent” – Is coarsened is the correct word here? Or did you calculate σ_w over 50km grid cells using the 6km w (or obs) data?